# Glial reactivity and cognitive decline follow chronic heterochromatin loss in neurons

A. G. Newman [1] ✉, J. Sharif [2], P. Bessa [1], S. Zaqout [3], J. P. Brown[1], D. Richter[1], R. Dannenberg[1], M. Nakayama [4], S. Mueller [5,6], T. Schaub[1], S. Manickaraj[1], P. Boehm-Sturm [5,6,7], O. Ohara [8], H. Koseki [2], P. B. Singh [1,9,11] ✉ & V. Tarabykin [1,10,11] ✉

In aging cells and animal models of premature aging, heterochromatin loss coincides with transcriptional disruption including the activation of normally silenced endogenous retroviruses (ERVs). Here we show that loss of heterochromatin maintenance and de-repression of ERVs result in a chronic inflammatory environment characterized by neurodegeneration and cognitive decline in mice. We identify distinct roles for HP1 proteins to ERV silencing where HP1γ is necessary and sufficient for H4K20me3 deposition and HP1β deficiency causes aberrant DNA methylation. Combined loss of HP1β and HP1γ results in loss of DNA methylation at ERVK elements. Progressive ERV de-repression in HP1β/γ DKO mice is followed by stimulation of the integrated stress response, an increase of Complement 3+ reactive astrocytes and phagocytic microglia. This chronic inflammatory state coincides with age-dependent reductions in dendrite complexity and cognition. Our results demonstrate the importance of preventing loss of epigenetic maintenance that is necessary for protection of postmitotic neuronal genomes.

Aging neurons operate under conditions of high cellular stress without renewal by cell division. Under normal physiological conditions, neurons continuously break their own DNA at enhancer regions[1,2], and their high metabolic rate results in an excess of reactive oxygen species harmful to protein and DNA integrity[3]. Neuronal genes most sensitive to oxidative DNA damage are downregulated with age[4], resulting in either cell death or cascading dysfunction and de-differentiation[5].

Little is known about what occurs in heterochromatin in aging neurons. However, in other cell types and human progeria models, a loss of DNA methylation[6,7], histone 3 lysine 9 tri-methylation (H3K9me3) and associated proteins is observed with age[8–10], along with a decreased association of heterochromatin with the nuclear lamina[11]. These age-related changes result in the activation of normally silenced repetitive elements such as endogenous retroviruses (ERVs)[12] and Long Interspersed Nuclear Elements (LINEs)[10]. Elevated

[1]Institute of Cell and Neurobiology, Charité – Universitätsmedizin Berlin, Corporate member of Freie Universität Berlin and Humboldt-Universität zu Berlin, Berlin, Germany. [2]Developmental Genetics Laboratory, Center for Integrative Medical Sciences (IMS), RIKEN National Research and Development Agency, 1-7-22 Suehiuro-cho, Tsurumi-ku, Yokohama-shi, Kanagawa-ken, Japan. [3]Department of Basic Medical Sciences, College of Medicine, QU Health, Qatar University, Doha, Qatar. [4]Laboratory of Medical Omics Research, Department of Frontier Research and Development, Kazusa DNA Research Institute, 2-6-7 Kazusa-Kamatari, Kisarazu, Chiba, Japan. [5]Charité-Universitätsmedizin Berlin, Charité 3R – Replace | Reduce | Refine, Charitéplatz 1, Berlin, Germany. [6]Charité-Universitätsmedizin Berlin, Department of Experimental Neurology and Center for Stroke Research Berlin, Charitéplatz 1, Berlin, Germany. [7]Charité-Universitätsmedizin Berlin, Charité Core Facility Experimental MRIs, Charitéplatz 1, Berlin, Germany. [8]Department of Applied Genomics, Kazusa DNA Research Institute, 2-6-7 Kazusa-Kamatari, Kisarazu, Chiba 2, Japan. [9]Department of Biosciences, School of Medicine, Nazarbayev University, 5/1 Kerei, Zhanibek Khandar Street, Astana, Kazakhstan. [10]Institute of Neuroscience, Laboratory of Genetics of Brain Development, National Research Lobachevsky State University of Nizhny Novgorod, 603022 23 Gagarin Avenue, Nizhny Novgorod, Russia. [11]These authors contributed equally: P. B. Singh, V. Tarabykin. ✉e-mail: andrew.newman@pm.me; prim.singh@nu.edu.kz; tarabykinvictor@gmail.com

transcription of ERVs have been observed in pathological states such as exogenous viral infections[13,14], cancer[15], neurodegeneration[16,17], multiple sclerosis[18,19], ALS[20], and Alzheimer's disease[17]. Elevated levels of ERVs have also been seen in models investigating factors associated with neurodegenerative diseases such and Tau[21] and TDP-43[22], while α-synuclein has been shown to affect chromatin and the maintenance of ERVs directly[23,24]. However, a causal relationship between ERVs and the initiation of neurodegeneration has yet to be determined.

ERVs are silenced by the KAP1 repressor complex which recruits histone de-acetylases, DNA methyltransferases, and histone methyl-transferases and several cofactors to induce heterochromatin formation (reviewed in ref. [25]). Here, the histone methyltransferase SETDB1 catalyzes H3K9me3 methylation[26], which serves as a high-affinity binding site for Heterochromatin Protein 1 (HP1)[27,28], which facilitates compaction and silencing[29].

We mimicked age-related heterochromatin loss by deletions of members of the HP1 family in the mouse brain. Unlike other mutants of enzymatic epigenetic modifiers—which typically have severe developmental phenotypes—the removal of HP1 proteins mimics the destabilization normally seen in aged cells, whose lower levels of H3K9me3 naturally result in less HP1 binding, activity, and stability. In doing so, we describe the molecular contributions of HP1β (*Cbx1*) and HP1γ (*Cbx3*) to ERV silencing and uncover an endogenous cause of the known age-related increases[30,31] of Complement in the brain.

All three HP1 homologs are robustly expressed in post-mitotic neurons (Figure S1a, b). To test whether heterochromatin loss can drive neuronal aging in vivo, we engineered mice to be conditionally deficient for HP1β and HP1γ in the cerebral cortex using the *Emx1^Cre* deleter mouse strain. We observed a partial malformation of the infrapyramidal blade of the dentate gyrus (DG) due to depletion of Ki67+ progenitors following loss of HP1β (Figure S1c,d), which is consistent with HP1β's established role in mitotic stability[32]. Apart from this minor developmental defect, we detected no overt changes in cortical cytoarchitecture in HP1 single and double mutants (Figure S1e, f), making these mutants a suitable model to study the long-term effects of HP1-related heterochromatin loss.

## Results

### HP1 deficiency results in de-repression of ERVs and an innate immune response

Given the known role of HP1 proteins in maintaining repression of non-coding elements[33], we performed cRNA-RNA in situ hybridization for the murine endogenous retrovirus (ERV) Intracisternal Alpha Particle (IAP) and could observe its robust de-repression in the HP1β/γ DKO, which was especially strong in the hippocampus and absent from the dentate gyrus (Fig. 1 and Figure S2b). Analysis of RNAseq from young and aged hippocampi using TETranscripts, where genes are quantified alongside aggregate counts of repeat elements (Fig. 1a, Supplementary Data 1) confirmed relatively few changes in gene transcription alongside robust upregulation of primarily ERVK- derived repeats such as IAP elements. When looking at locus-specific changes in repeat expression, we could observe some loci showing de-repression after single loss of HP1β or HP1γ, such as an IAPez element on chromosome 7, genome wide de-repression occurs in HP1β/γ DKO, with 287 loci being significantly (log2FC > 1, padj < 0.05) affected (Fig. 1b, and Supplementary Data 2). The repeats affected correspond primarily to evolutionarily recent elements, evidenced by a very small kimura distance (Fig. 1c). De-repression of these recent mouse-specific elements also resulted in elevated chimeric transcription in HP1β/γ DKOs (S2c,d & Supplementary Data 1). At the genic level, genotype-dependent variance fell into three categories (Fig. 1a, and Supplementary Data 1): The first are dentate gyrus related genes (140) such as Prox1, Dsp, Trpc6, Plk5 and Cdh9 that are underrepresented in HP1β KO and HP1β/γ DKO (Supplementary Data 1). The remaining two categories were upregulated (549): first, expression of the entire clustered

protocadherin (cPcdh) locus was elevated in HP1γKO and HP1β/γ DKO (which is further observed in gene ontology analyses (Figure S2e)), as were a small subset of canonical and non-canonical imprinted genes (Supplementary Data 1). Second, after a Gene Set Enrichment Analysis of HP1β/γ DKO transcriptomes (c2.cp.reactome.v6.2) we observed changes in genes related to inflammation and integrated stress response that include innate immune pathways, Toll receptor signaling, the Unfolded Protein Response (UPR), MHC class I and II antigen presentation, and the Complement Cascade (Fig. 1e, and Figure S2f–g).

### HP1γ is required for H4K20me3 deposition and regulation of Protocadherin genes

To understand the epigenetic pathway by which HP1β and HP1γ regulate repression of ERVs, we investigated the effect of HP1β and HP1γ mutations on the HP1-related histone modifications H3K9me3 (histone 3 lysine 9 trimethylation) and H4K20me3 (histone 4 lysine 20 tri-methylation). H3K9me3-bound HP1 can recruit Suv420h1/2 HMTases and direct local H4K20me3 deposition[34], and both histone modifications are abundant in wt post-mitotic neurons (Fig. 2a, and Figure S3a). We observed a specific loss of H4K20me3 in HP1γ-deficient neurons (Fig. 2a, and Figure S3b), consistent with previous observations in spermatocytes[35]; H3K9me3 levels were unaffected. We also found that HP1γ was sufficient for H4K20me3 deposition because re-addition of HP1γ to HP1β/γDKO cortices by in utero electroporation at E14 restored H4K20me3, albeit not to the levels seen in adjacent interneurons where HP1γ is not deleted (Fig. 2b). The H3K9me3-HP1γ-H4K20me3 pathway appears to regulate isoform selection at the cPcdh locus. While all protocadherin isoforms can be observed in bulk RNAseq, single neurons express a unique combination of protocadherin isoforms which is clonally defined during neurogenesis[36–38]. Thus, bulk ChIPseq shows the protocadherin cluster is marked with H3K9me3 and H4K20me3 (Figure S3c) corresponding to single cell silencing of unused exons observed at the population level. In the HP1γKO, H4K20me3 is lost and there is elevated expression of cPcdh exons at the population level (Fig. 1a, Figure S3c), indicating a loss of exon exclusion in single cells. The exact consequence for protocadherin isoform specificity at the single cell level remains to be explored. Given the known requirement of the HP1 chromoshadow domain (CSD) for association with the H4K20me3 HMTase Suv420h2[39], we carried out co-immunoprecipitation and co-localization analysis to identify residues in the CSD essential for the interaction of HP1γ with Suv420h2. We observed that mutation of I165, F167, L172 in addition to loss of the entire CSD inhibited the interaction between HP1γ and Suv420h2 (Fig. 2c–e, Full length blots are in Source Data).

### HP1β deficiency results in aberrant DNA methylation

Prominent increases in the expression of tissue specific imprinted genes (Supplementary Data 1) in HP1βKO and HP1β/γ DKO mutants indicated that DNA methylation may be affected by HP1β deficiency. Prompted by these observations, we engineered an ES cell line that contains an ERT2-Cre transgene where all three HP1 genes are floxed. This system allowed for the deletion of all three HP1 genes following the addition of tamoxifen, (hereafter termed HP1cTKO) that we then profiled for DNA methylation using reduced-read bisulfite sequencing (RRBS) and changes to KAP1, H3K9me3 and H4K20me3 (Fig. 3a). We found that triple deficiency of HP1 proteins results in reduced KAP1 at imprinting control regions (ICRs) marked by ZFP57 (Figure S4a), while DNA methylation at these ICRs including *Nnat* shows mixed changes (Figure S4b, c). KAP1 was unchanged over IAP elements in HP1cTKO ES cells, with a comparative reduction in H3K9me3 and an expected absence of H4K20me3 (Fig. 3b). We acknowledge that these ChIP-seq results did not have input normalization but have been normalized using 1 × RPGC (Reads Per Genomic Content) to account for sequencing depth and genome mapability. While ES cells are not expected to utilize protocadherins the same way as neurons, regulatory H3K9me3

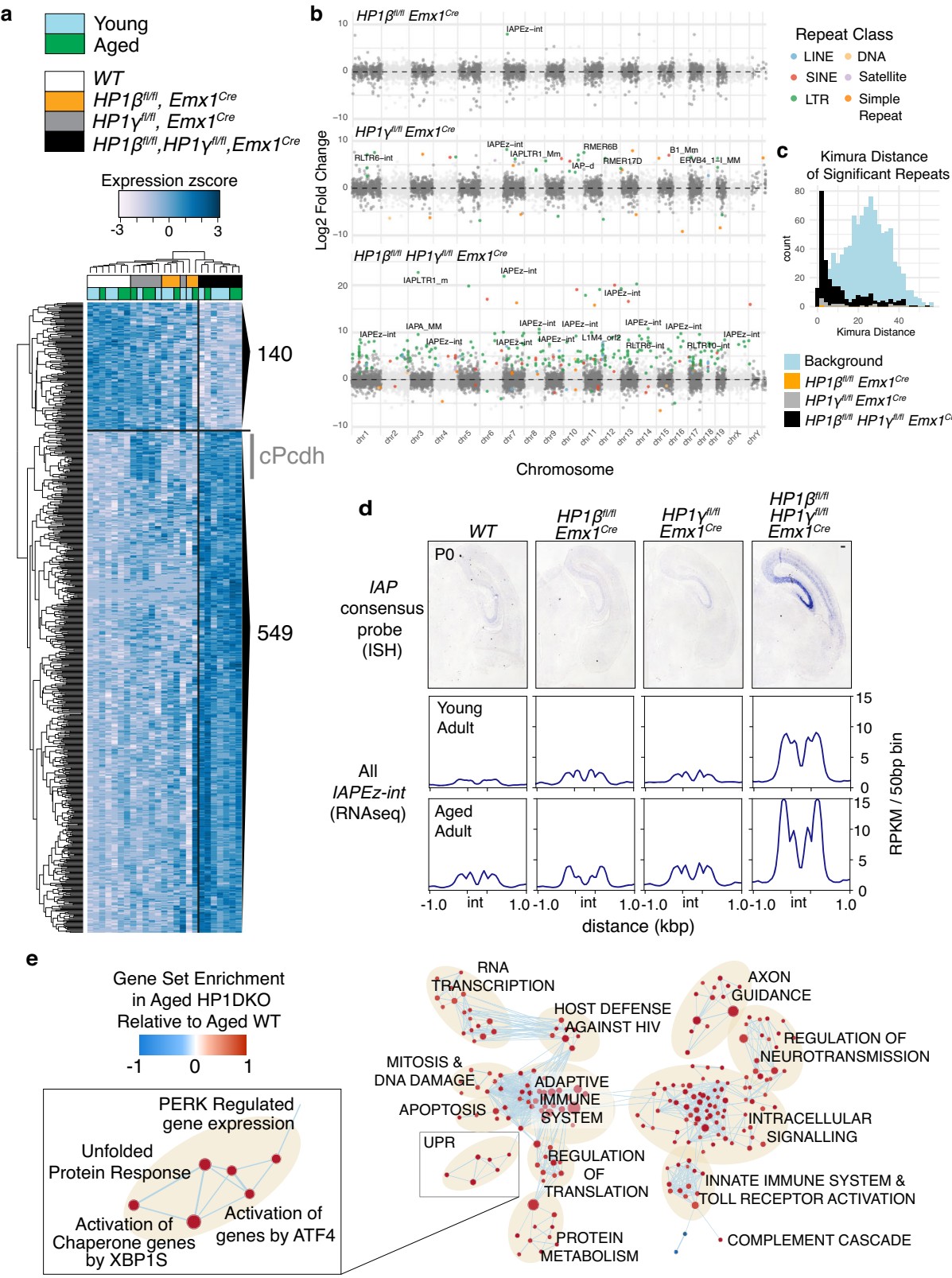

**a** Young / Aged; WT; HP1β^{fl/fl}, Emx1^{Cre}; HP1γ^{fl/fl}, Emx1^{Cre}; HP1β^{fl/fl}, HP1γ^{fl/fl}, Emx1^{Cre}; Expression zscore −3 0 3; 140; cPcdh; 549

**b** HP1β^{fl/fl} Emx1^{Cre}; HP1γ^{fl/fl} Emx1^{Cre}; HP1β^{fl/fl} HP1γ^{fl/fl} Emx1^{Cre}; Log2 Fold Change; Chromosome; Repeat Class: LINE, DNA, SINE, Satellite, LTR, Simple Repeat

**c** Kimura Distance of Significant Repeats; count; Kimura Distance; Background; HP1β^{fl/fl} Emx1^{Cre}; HP1γ^{fl/fl} Emx1^{Cre}; HP1β^{fl/fl} HP1γ^{fl/fl} Emx1^{Cre}

**d** WT; HP1β^{fl/fl} Emx1^{Cre}; HP1γ^{fl/fl} Emx1^{Cre}; HP1β^{fl/fl} HP1γ^{fl/fl} Emx1^{Cre}; IAP consensus probe (ISH); P0; All IAPEz-int (RNAseq); Young Adult; Aged Adult; RPKM / 50bp bin; distance (kbp)

**e** Gene Set Enrichment in Aged HP1DKO Relative to Aged WT; −1 0 1; PERK Regulated gene expression; Unfolded Protein Response; Activation of Chaperone genes by XBP1S; Activation of genes by ATF4; UPR; RNA TRANSCRIPTION; HOST DEFENSE AGAINST HIV; MITOSIS & DNA DAMAGE; APOPTOSIS; ADAPTIVE IMMUNE SYSTEM; REGULATION OF TRANSLATION; PROTEIN METABOLISM; AXON GUIDANCE; REGULATION OF NEUROTRANSMISSION; INTRACELLULAR SIGNALLING; INNATE IMMUNE SYSTEM & TOLL RECEPTOR ACTIVATION; COMPLEMENT CASCADE

in the cPcdh is unaffected in HP1cTKO ESCs despite H3K9me3 and H4K20me3 being lost at adjacent IAP elements (Figure S4d, arrows). This additionally suggests that regulatory H3K9me3 is unaffected in *HP1γ^{fl/fl} Emx1^{Cre}* mutants, and that it is disruption of the HP1γ-H4K20me3 pathway that causes elevated cPcdh expression in HP1γKO and HP1β/γDKO brains.

DNA methylation can be used to accurately estimate the biological or 'epigenetic' age (eAge) of tissues, which increases at a steady rate in adult tissues[7]. While embryonic stem cells normally maintain an assigned eAge of zero[7], we were surprised to find that the eAge of HP1cTKO ES cells increased faster than their time in culture (Fig. 3c).

**Fig. 1 | Loss of both HP1β and HP1γ results in activation of noncoding elements and induction of the integrated stress response. a** Genes (inc. aggregate counts of repetitive elements measured using TETranscripts) significantly changed in young and aged HP1β/γDKO hippocampi (689 genes, edgeR quasi-likelihood F-test, corrected $p < 0.05$). **b** Manhattan plot of loci specific significant changes (SalmonTE*[101],) to repetitive element transcription in HP1 mutants. Significantly changed repeats (DESeq2 negative binomial generalized linear model, Wald test, negative binomial generalized linear model, adj $p < 0.05$ & log2FC > 1) are colored by repeat class. **c** Kimura Distance distribution of Repeats significantly changed in HP1 mutants. Repeats from the ERVK subfamily, which contains the evolutionarily recent *IAP*s, is most strongly affected in HP1β/γDKO hippocampi, which is reflected in a very small Kimura distance. Here the background distribution is a 1/1000th downsampling of all repeats in the dataset. **d** In situ *hybridization* using a consensus probe for IAP in P0 brains (representative image shown for 3 attempted assays) and RNAseq read coverage over the IAPEz internal fragment in young and aged adults (read coverage is RPKM normalized reads per 50 bp bin). Scale bar = 500μm. **e** Enrichment map of significant reactome pathways observed from Gene Set Enrichment Analysis (GSEA) in aged HP1β/γDKO transcriptomes. Reactome pathways were filtered to FDR < 0.25 (Figure S2f, Supplementary Data 3).

HP1cTKO ESCs displayed DNA hypomethylation at LINEs and LTRs (particularly ERVK) (Fig. 3d, g) despite a global background shift towards DNA hypermethylation (Fig.3e) which is also observed at promoters and gene bodies (Fig. 3f). While many significantly hypermethylated CpGs overlap with CpG islands, exons, SINEs (Fig. 3d) and ICRs (Figure S4b), as annotation sets they do not show statistical significance based on the hypergeometric test (Fig. 3g), which may be attributed to the global background shift towards hypermethylation.

Given that reduced representation bisulfite sequencing (RRBS) cannot discriminate between 5-methyl cytosine (5mC) and 5-hydroxymethyl cytosine (5hmC), we hypothesized that some of the 'hypermethylation' observed in the HP1cTKO RRBS samples may be attributable to active demethylation processes, specifically the oxidation of 5mC to 5hmC mediated by TET enzymes[40]. We tested two specific the *Nnat* locus and IAP elements using primers designed to amplify over single HpaII/MspI (CCGG) sites from hippocampal lysates. 5mC at Neuronatin (*Nnat*) is decreased in an age-dependent manner in *HP1β^{fl/fl} Emx1^{Cre}* hippocampal lysates. Similarly, there were HP1β/γDKO age-dependent decreases in 5mC at IAP sequences hippocampal lysates. There was trend towards increased 5hmC for both loci (Figure S5a).

To elucidate the full effects of HP1 deficiencies on 5mC and 5hmC, we conducted RRBS both with and without prior oxidation on hippocampal lysates. This oxidation step converts 5hmC into 5-formylcytosine (5fC), which does not undergo bisulfite conversion, thereby allowing us to distinguish between 5mC and 5hmC (Fig. 4a). After an average 67% unique ( + 20% ambiguous) read mapping alignment efficiency we found that bisulfite conversion using this method yielded on average ~35% modified cytosines (Fig. 4b). This experiment yielded results largely consistent with what was observed in HP1cTKO ESCs. We found that deficiency of HP1β results in aberrant 5mC (de novo) hypermethylation of promoters and CpG islands, a large percentage coming from a gene desert on chromosome 12 (Fig. 4c–e, and Figure S5c). Surprisingly, we also found that clustered protocadherin promoters became dramatically 5mC hypomethylated in HP1β/γDKO hippocampi (Fig. 4e, and Figure S5d)., along with imprinting control regions (ZFP57 ICRs) (Fig. 4e). 5mC DNA methylation at LTRs, primarily ERVK elements, was markedly reduced in HP1β/γDKO hippocampi (Fig. 4e, and Figure S5e). This effect is also likely an underestimate given the lower mapability of repeats in RRBS sequencing and a lower bisulfite conversion efficiency in this experiment. We could also observe corresponding statistically significant increases in 5hmC over ERVK and IAPLTR1a annotated regions in HP1β/γDKOs. Notably, single deficiency of either HP1β or HP1γ already initiates drift in DNA methylation, evidenced by significant increases in 5hmC over introns, LINEs, SINEs, and LTRs, an effect that is recapitulated in normal aging in wildtype hippocampi (Fig. 4e).

## Activation of astrocytes and microglia

We found that 15% of (88/565 protein coding) genes differentially expressed in HP1β/γDKO hippocampi overlapped with cellular responses to interferons including *Ifitm2, Ifi27* and the regulatory component of the interferon gamma receptor *Ifngr2* (Supplementary Data 1). HP1β/γDKO hippocampi also showed increased *Oas3, Lyk6,*

*Wdfy4*, and *Il34*, but most notably displayed elevated transcription of complement *C3, C4b* and *C1qa*.

Given the recently established importance of complement proteins in the developing brain[41], their co-occurrence with amyloid plaques[42], and their accumulation during normal aging[30,43], we performed a multiplex in situ hybridization in order to identify the cell type(s) containing raised levels of Complement 3 (*C3*) RNA (Fig. 5a–i). Elevated *C3* transcripts could be observed in the soma of wildtype CA1 and CA3 neurons, but in aged HP1β/γ DKO, *C3* could also be detected in small plaque-like foci in the *stratum radiatum* (white arrows, Fig. 5d, g) that were not observed in any other condition. These foci were surrounded by Iba1+ microglia with a distinct morphology (Fig. 5e, f). A second experiment revealed these *C3*+ foci were also *Slc1a3*+ indicating these foci were reactive astrocytes (Fig. 5g–i and Figure S6a, b). In aged HP1β/γ DKO hippocampi, the number of total GFAP+ astrocytes significantly increases compared to young DKO animals, although this puts it in the same range as the other genotypes. Notably, ~50% more Iba1+ microglia can be observed in aged HP1β/γ DKO hippocampi (Fig. 5j, k and Figure S6c, d), including significant migration into the 75μm range where the soma of hippocampal pyramids are located (Figure S6c, d). Iba1+ microglia neighboring GFAP+ astrocytic foci in the *stratum radiatum* of HP1DKO hippocampi exhibited large CD68+ protrusions (arrows Fig. 5l), indicating augmented phagocytosis. We quantified the CD68+ area within Iba1+ cells and found that this significantly increased in HP1β/γ DKO hippocampi in an age-dependent manner (Fig. 5m). This suggested that pro-inflammatory signalling, likely because of chronic de-repression of IAPs and other ERVs, results in greater activation of microglia. We could further localize complement C3 signal in aged hippocampi using immunohistochemistry. While aged wildtype hippocampi show minor C3 signal around blood vessels (Fig. 5n, o), C3 signal is observed sporadically in astrocytes in aged HP1βKO and HP1γKO animals (Fig. 5p-s), which may be due to minor de-repression of ERVs in single mutants (Fig. 1b). By contrast, HP1β/γ DKO hippocampi show an abundance of C3 protein in astrocytes, which is primarily observed at astrocytic end feet (Fig. 5u, white arrows). We did not observe de-repression of ERVs in astrocytes in HP1β/γ DKO mutants (Fig. 5d, f, and Figure S7). Rather, we observed preferential ERV de-repression in deep layer neurons of the cortex and hippocampal pyramids but not the dentate gyrus (Figs. 1d, and S2b, 5). This suggests both that ERV re-activation requires cell-type specific transcription factors (such as previously suggested Sox-family transcription factors[44]), and that inflammatory effects on astrocytes and microglia are non cell-autonomous.

Given that IAP transcripts de-repressed in HP1β/γ DKO brains are neuronally derived and glial activation follows, we designed an experiment to profile the stimulatory effect of IAP ssRNA on mixed glial cultures in vitro (Fig. 6a, b). As a control, we used a scrambled ssRNA of equivalent length and as a second control, we used the identical IAP sequence but substituted with pseudo uracil[45], thus termed ψ-IAP (Fig. 6b). 24 h following introduction of IAP ssRNA we profiled the media and cytosolic lysate for activation of chemokines and cytokines using a membrane-based sandwich immunoassay (Fig. 6c). scrRNA was mildly immunogenic in contrast to IAP ssRNA

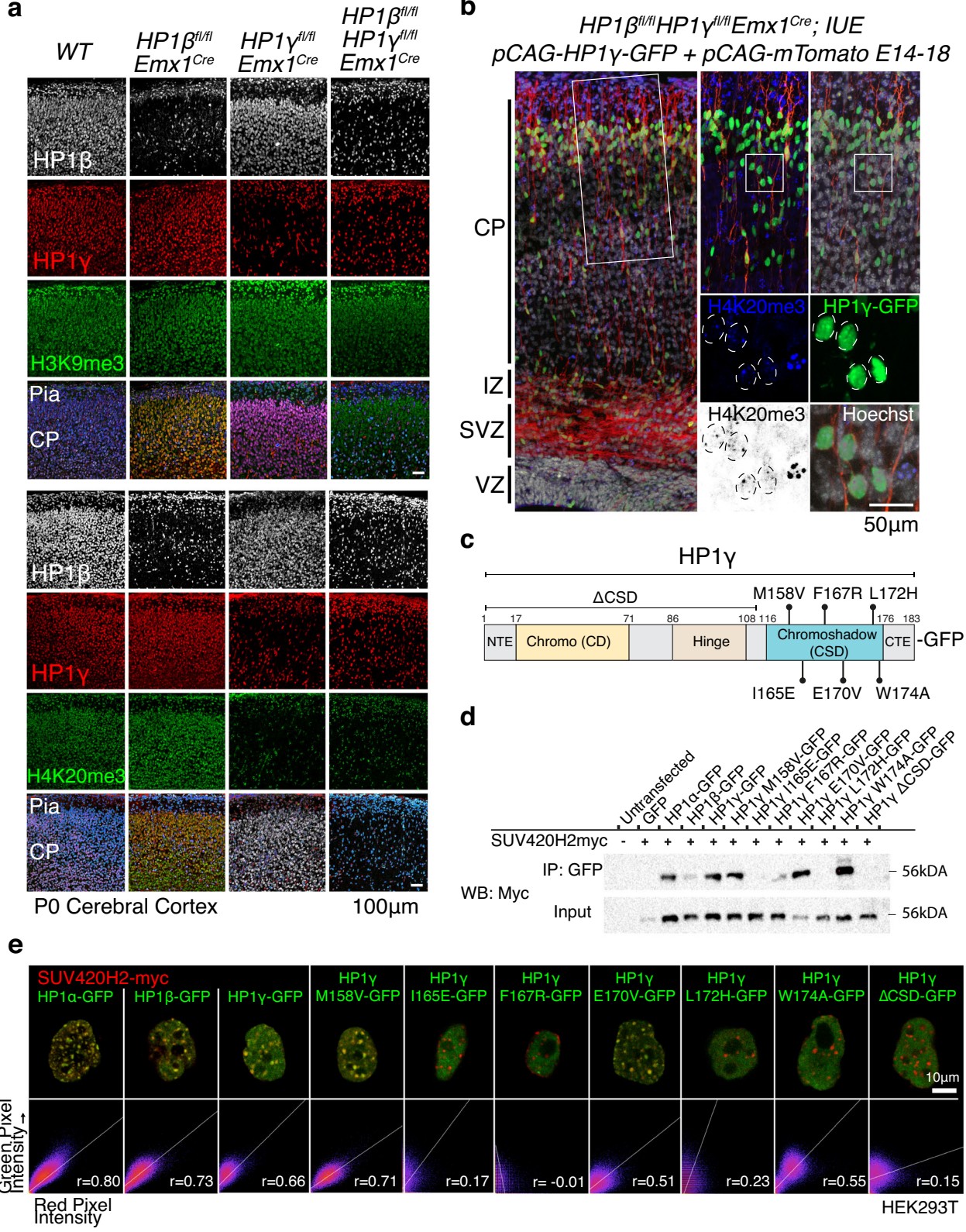

where we observed specific activation of several classical cytokines in response to IAP ssRNA including CXCL10, IL-23, IL-1ra and CCL5 (RANTES) (Fig. 6c). CCL5 activation came from GFAP+ astrocytes predominantly and was specific to IAP ssRNA and not ψ-IAP ssRNA (Fig. 6d–f). We also tested the response to IAP ssRNA on interferon response element 7 (IRF7) and found that IAP ssRNA greatly promoted the higher molecular weight band, corresponding to activated,

phosphorylated form that enters the nucleus[46]. This was confirmed by immunohistochemistry, where astrocytes strongly responding to IAP ssRNA with CCL5 activation showed increased nuclear IRF7 (Fig. 6g). Neither our 24 hr or 48 hr paradigms could detect changes in cytoplasmic C3 by Western blot (**see Supplemental Information**), however, our acute experimental protocol is unlikely to fully recapitulate the chronic inflammatory environment in the HP1β/γDKO brain.

**Fig. 2 | HP1γ is necessary and sufficient for deposition of H4K20me3.**
**a** H3K9me3 levels are unchanged in HP1β and HP1γ deficient neurons. H4K20me3, is unaffected in wild-type interneurons but lost completely in HP1γ-deficient pyramidal neurons. Representative images are shown from 3 attempted assays. **b** Re-addition of HP1γ into developing mutant brains by *in utero electroporation* can restore H4K20me3 (magnification, nuclei outlined by dashed circles). (CP = cortical plate IZ = intermediate zone, SVZ = subventricular zone, VZ = ventricular zone). Representative images are shown from 3 attempted assays. **c** Schematic of HP1γ

protein product with domains and point mutations tested. **d** Co-immunoprecipitation of Suv420h2 with HP1 proteins and point mutants confirms residues in the chromoshadow domain (CSD) of HP1γ are essential for its binding with Suv420h2. Representative blots are shown from 2 attempted assays, reverse immunoprecipitation can be found in Supplemental information. **e** Co-localization of C terminal GFP-tagged HP1α, HP1β, HP1γ and HP1γ mutants with C terminal myc-tagged SUV420H2. Representative images are shown from 2 attempted assays.

## Increased dendritic loss and cognitive decline

Young HP1β KO CA3 hippocampal neurons showed a modest reduction in CA3 basal dendrite complexity that was carried through to aged animals. By contrast, HP1β/γ DKO animals displayed a pronounced age-dependent decrease in CA3 basal dendrite complexity (Fig. 7a). Structural MRI revealed that young HP1β/γ DKO animals have a modestly reduced volume of both the cortex and total brain, albeit the difference is less obvious in aged HP1β/γ DKO animals (Figure S8).

Cognition and behavior of HP1 mutant mice was profiled as young adults (3–4 months) and again in middle age (13–14 months), where we observed several age-dependent deficits. HP1β/γ DKO animals exhibited deficits in spatial learning and memory as tested in the Barnes maze (Fig. 7b). Young HP1β/γ DKO animals showed impairment in both learning (24 h test) and recall (7 day test) of the target nest, while aged HP1β/γ DKO animals seemed unable to learn the location of the target nest and typically walked around the periphery continuously. Aged HP1β/γ DKO animals also showed an age-dependent abolition of paired-pulse inhibition (Figure S9a), extended bouts of eating and grooming (Figure S9b) and an altered circadian rhythm (Figure S9f). During handling and testing, a subset of HP1 mutant animals displayed stimulus-dependent seizures, which were observed with HP1β KO animals predominantly (Figure S9e). Young HP1β/γ DKO animals displayed hyperactivity in an open field, and as they aged showed both an absence of center zone anxiety (Figure S9c) and a marked inability for nest construction (Figure S9d).

## Discussion

We have observed a role for HP1 proteins in regulating DNA and histone modifications in vivo. Age-dependent reduction in DNA methylation at the imprinted *Nnat* gene in HP1βKO indicates that HP1β stimulates Dnmt1 activity, as described previously[47]. That Dnmt1 and HP1β operate along the same pathway is also indicated by the similar phenotypic effects of HP1βKO on dentate gyrus formation compared to *Nestin-Cre* deletion of *Dnmt1*[48]. Further, robust activation of IAP elements in the HP1β/γDKO mutant phenocopies both the *KAP1^{fl/fl}Emx1^{Cre}*[49] and *Dnmt1^{fl/fl}Emx1^{Cre}* mutants, absent the malformation of the cortex in *Dnmt1^{fl/fl}Emx1^{Cre}* mutants[50], which we did not observe. Notably, many of the genes upregulated in HP1β/γDKO hippocampi overlap with regions where Dnmt1-dependent methylation is not recovered once abolished[51]. Since 5mC methylation is dramatically lost at ERVK elements in HP1β/γDKO but not in single mutants, this suggests that *HP1β or HP1γ* are capable of stimulating Dnmt1 activity and the loss of both is detrimental. Age-related 5mC hypomethylation appears to be restricted to loci already under threat of activation such as the tissue-specific imprinted genes (*i.e., Nnat*) or partially silenced ERVK repeats such as IAPs. This phenomena has been observed before with IAP elements in aging mice—where their periodic activation results in progressive demethylation and complete de-silencing[52], and may explain why IAP transcription further increases in aged HP1β/γDKO (Fig. 1c). The shift towards hydroxymethylation in HP1β/γDKO hippocampi along with the measured increases in age-related hydroxymethylation in non-coding regions (Fig. 4e) indicate that HP1 proteins also serve to protect against TET-mediated hydroxymethylation.

HP1γ function is required for deposition of H4K20me3 at the cPcdh cluster. cPcdh is known to be regulated by the SETDB1 HMTase that generates H3K9me3[53] and it seems likely that a H3K9me3-HP1γ-

H4K20me3 pathway might be an important mechanism for regulating chromosomal domains, such as the cPcdh cluster and clustered retrotransposons in general.

HP1β/γDKO hippocampi share several molecular characteristics with very aged (24-29 month) mouse hippocampi. For one, like very aged hippocampi, HP1β/γDKO hippocampi display downregulation of ZFP57, Suv420h2, and upregulation of much of the protocadherin cluster[54]. Further, some of the strongest transcriptional changes in HP1β/γDKO hippocampi are also the strongest changes observed in very aged hippocampi, including age-associated ncRNAs *Pisd-ps1* and *Pisd-ps2* and complement components *C1qa, C3,* and *C4b*[43,54–56].

HP1β/γDKO hippocampi exhibit a 'pre-plaque' state similar to that found in very old hippocampi and in early Alzheimer's disease, where complement seeding, microglial activation and synapse loss precede plaque pathology[57]. It has been suggested that the mechanism by which astrocytes and microglia co-ordinate the complement cascade involves the initial activation of microglia. Activated microglia then secrete Il-1a, TNF and C1q that induce 'A1'-astrocyte reactivity[58]. In addition to being directly neurotoxic[58], it is thought reactive astrocytes also greatly enhance the susceptibility of aging brains to neurodegeneration because they are major source of classical complement cascade components C3 and C4b that then drive sustained microglia-mediated synapse loss[57]. This sustained inflammation maintains reactive astrocytes and complement[31], both of which have long been associated with senile amyloid plaques (for review see ref. 59). Alongside immune and complement responses, HP1β/γDKO hippocampi show enrichment in semaphorin signaling pathways (Figure S2g, and Supplementary Data 3), which we attribute to known semaphorin signatures in microglia-astrocyte communication during reactive chemotaxis[60,61].

Although it is unclear how microglia are activated in HP1β/γDKO hippocampi, a likely sequence of events is that microglia activation is a response to the export of ERV or other inflammatory RNA from HP1β/γDKO neurons that is detected by astrocytes and microglia. Export of RNA or protein-aggregate filled extracellular vesicles (EVs) have been observed in neuron-neuron and neuron-glia communication[62,63], and ERVs have been shown to be included in EVs[64]. In this context, human ERVs have been found to be elevated in Alzheimer's[17] and ALS[20]. Extracellular vesicles that export neuronal unfolded protein aggregates have also been shown to serve as an activating signal to microglia or astrocytes[63] and this is sufficient to drive non-cell-autonomous neuronal degeneration[65]. We note that while C3 RNA appears to be produced by both neurons and astrocytes in HP1β/γDKO (Fig. 5g–i, and Figure S6b), C3 protein is detected primarily at astrocytic endfeet (Fig. 5u). In addition to contacting neurons, astrocytic endfeet are known to contact endothelial cells and pericytes of the blood brain barrier, and it is possible that the complement response to ERV RNA in the HP1β/γDKO brain is the result of a greater systemic immune response.

When we tested the effect of non cell-autonomous ERV RNA on mixed glial cultures, we could observe some ERV RNA specific effects. The acute ERV ssRNA stimulation was followed by IRF7 phosphorylation and nuclear ingress (Fig. 6e-g). IRF7 activation occurs downstream of Pattern Recognition Receptors (PRRs) that recognize ssRNA, either endosomal toll receptors TLR7/9, or cytoplasmic receptors such as RIG-1/MDA5[66]. Once in the nucleus, phosphorylated IRF7 is known to

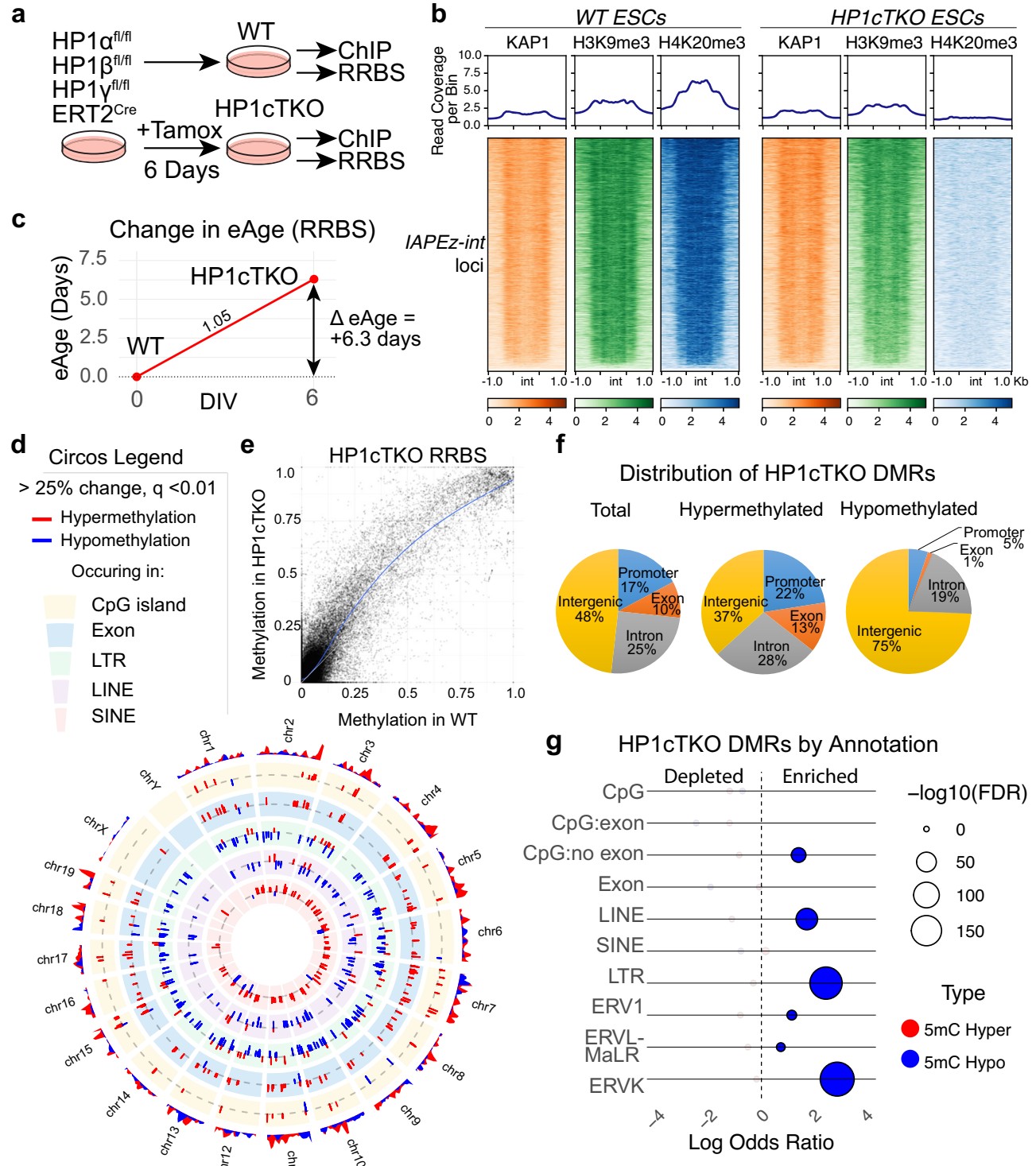

**Fig. 3 | DNA methylation is further perturbed in HP1cTKO ES cells. a** Schematic of the HP1cTKO RRBS experiment where $HP1\alpha^{fl/fl}HP1\beta^{fl/fl}HP1\gamma^{fl/fl}ERT2^{Cre}$ ES cells are left untreated (WT) or treated with tamoxifen (HP1cTKO) for 6 days in vitro before profiling DNA methylation status by Reduced Representation Bisulphite Sequencing (RRBS) and H3K9me3, H4K20me3 and KAP1 by ChIPseq. **b** Coverage of KAP1, H3K9me3 and H4K20me3 over *IAPEz internal segments (int)* and adjacent LTRs is reduced in HP1cTKO ES cells. Given the deletion of HP1γ in HP1cTKO, H4K20me3 is lost entirely. **c** Profiling of -18,000 CpG sites in WT and HP1cTKO ES cells reveals that deletion of HP1 proteins initiates a positive change in eAge. **d** Circos plot of methylation changes that are greater than 25% (*q* < 0.01) plotted by chromosome by annotation. The magnitude of change is represented on the radial y axis. The

outermost ring represents methylation change density. Inner rings annotate methylation changes occurring to CpG islands, exons, LTRs, LINEs and SINEs respectively. **e** Scatterplot of cytosine methylation observed in both HP1cTKO and WT Reduced Representation Bisulfite Sequencing. **f** Distribution of HP1cTKO Differentially Methylated Regions (DMRs) by genic feature. **g** Odds ratios of HP1cTKO DMRs significantly changed (q < 0.05, > 25% change) plotted against the adjusted *P* value (FDR) of the respective hypergeometric test of DMRs overlapping with the annotation. CpG = CpG island, CpG:exon is the subset of CpG islands overlapping with exons, whereas CpG:no exon are CpG islands that do not overlap. ERV1, ERVL-MaLR and ERVK are subsets of LTR.

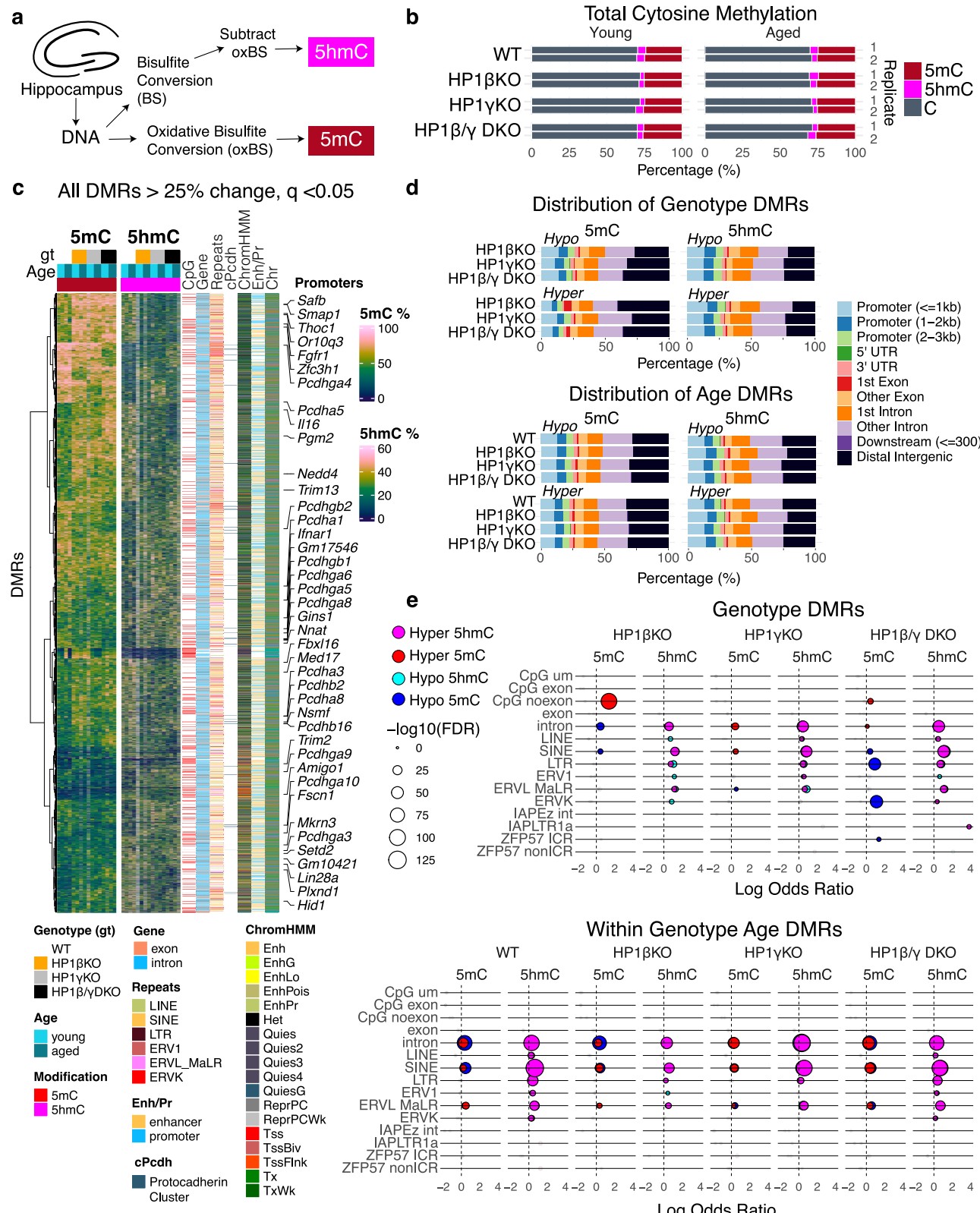

directly activate type I interferons (IFNα/β) and Interferon Stimulatory Genes (ISGs)[67], including the cytokines CXCL10 and CCL5[46], which we robustly observe 24hrs and 48hrs post ERV stimulation (Fig. 6). Both CCL5, along with the IL-23 receptor (IL-23R) have both been observed to be circulating biomarkers of senescence[68]. Collectively, these responses suggest that acute ERV stimulation induces a response converging on type I interferon response.

We did not observe changes to IFN-γ (type II interferon) in our acute paradigm, although the chronic environment of the HP1β/γDKO brain shows several pathways that point to activation of IFN-γ and a type II interferon response. Gene set enrichment analysis of HP1β/γDKO RNAseq shows increases for MHC class I and II antigen presentation pathways (Figure S2f, and Supplementary Data 3), known to be potently activated by IFN-γ[69]. We also note the upregulation of

**Fig. 4 | DNA methylation fidelity is progressively compromised in HP1 deficient hippocampi. a** Schematic of experimental design showing determination of paired 5-methyl Cytosine (5mC) and 5-hydroxy-methyl Cytosine (5hmC) measurements from young and aged HP1βKO, HP1γKO and HP1β/γDKO mutant hippocampal lysates by Reduced Representation Bisulfite Sequencing (RRBS). Genotypes that follow are abbreviated to HP1βKO, HP1γKO, and HP1β/γDKO respectively for clarity. **b** Global 5mC and 5hmC methylation across biological replicates sampled. **c** Heatmap of all observed Differentially Methylated Regions (5mC or 5hmC) that change 25% or more and are statistically significant below $q = 0.05$. DMRs (cytosine positions, rows) observed across genotypes are accompanied by row annotations corresponding to genomic context denoted by being a CpG island, in a gene (intron/exon), repetitive element (LINE, SINE, ERV1, ERVL_MaLR, ERVK or other nonredundant LTRs), overlap with the protocadherin cluster (denoted cPcdh), its chromatin state defined by the 18 state ChromHMM model from P0 mouse cortex (ChromHMM), its overlap with promoters or enhancers defined by the Enhancer-gene map from ENCODE 3, and chromosome (Chr). **d** Total distribution of DMRs by genotype and direction of change (hypomethylation 'hypo' or hypermethylation 'hyper'). **e** Odds Ratios of DMRs significantly changed due to genotype or age are plotted against the $Q$ value (FDR) resulting from the hypergeometric test of DMRs overlapping with the annotation. CpG um = Unmasked CpGs, CpG exon = CpG islands overlapping with exons, CpG noexon = CpG islands not overlapping with exons, LTR annotation here refers to all LTRs including ERV1, ERVL MaLR, ERVK etc.

IFNGR2 in HP1β/γDKO hippocampi (Supplementary Data 1), the regulatory subunit[70] of the interferon gamma receptor, suggesting greater sensitization to type II interferon signaling. In a chronic environment this may be especially problematic, given IFN-γ stimulation has been found to potentiate immune responses by initiating transcription of a subset of ERV dsRNA, which in turn triggers additional RIG-1/MDA5 signaling[71]. Astrocytes themselves seem to be highly sensitive to IFN-γ, with effects ranging from protective responses to cytotoxicity[72]. Activated astrocytes in many neurodegenerative diseases show upregulation of IFNGRs[73], suggesting an increase in sensitivity to IFN-γ signaling. Once effected, IFN-γ signaling has been observed to upregulate expression of the complement components C3 and C4 by stabilization of their mRNA[74,75]. Notably, in addition to other non-inflammatory roles, IFN-γ signaling has been shown to promote tau hyperphosphorylation[76]. In summary, we have observed that acute paradigms of ERV stimulation can robustly activate cytokines consistent with a type I interferon response, the chronic environment of the HP1β/γDKO brain suggests chronic activation of type II interferon response, which has several known synergies with complement signaling.

Our results add to a growing body of work implicating reactive RNA species, complement and unfolded protein response (UPR) engagement in neurodegenerative disease; cross-talk between innate immune pathways and the unfolded protein response (UPR) is known collectively as the integrated stress response (ISR)[77]. It is also known that extended UPR stress in astrocytes results in Complement activation[65] Complement is known to be activated in TDP-43 proteinopathies[78], which can be potentiated by elevated ERV expression[22]. Complement C1q also forms condensates in an RNA dependent manner that can drive neurodegeneration[79], which is consistent with what we observe, namely that astrocytes in HP1β/γDKO hippocampi accumulate C3 in an environment rich in ERV RNA (Fig. 5). UPR engagement in HP1β/γDKO hippocampi is likely a direct result of sustained ERV transcription (Fig. 1e, and Figure S2f, g). Given the majority of neurodegenerative diseases are characterized by misfolded proteins or altered proteostasis[77,80], our study provides an invaluable insight into how heterochromatin loss with concomitant (re)activation of ERVs can drive core components of neurodegeneration[81] by stimulation of immune pathways and the integrated stress response.

## Methods

### Animals

All mouse experiments were carried out in compliance with German law approved by the State Office for Health and Social Affairs, Council in Berlin, Landesamt für Gesundheit und Soziales (LaGeSo) under permissions G0079/11, G0206/16, G0184/20. Animals were euthanized by lethal injection of pentobarbital and confirmed by cervical dislocation.

The **H**eterochromatin **P**rotein 1 **F**loxed **E**mx1**C**re (HP1FEC) mouse line was generated by generating floxed alleles of HP1β (*Cbx1*) and HP1γ (*Cbx3*) (background 129/C57BL6J) which were combined with the Emx1-IRES-Cre mouse (Jackson Laboratory). The HP1β (*Cbx1*) targeted

allele was generated by introduction of a targeting cassette into ES cells by homologous recombination that inserted loxP sites surrounding exons II and III. Successful integrations were detected by neomycin selection via an FRT flanked neomycin cassette inserted between exons III and IV. Following a cross to a flip-deleter mouse, the NeO cassette is removed by flippase-mediated recombination, giving rise to the $HP1\beta^{fl/fl}$ allele (Figure S10). HP1γ (*Cbx3*) was targeted using a similar strategy giving rise to the $HP1\gamma^{fl/fl}$ allele (Figure S11). ERV derepression due to HP1 deficiency is not sex dependent and to reduce covariates, only male mice were used for behavioral experiments and sequencing experiments.

### Eukaryotic cell lines

**HP1cTKO embryonic stem cell line.** ESCs possessing *Cbx1*, *Cbx3*, or *Cbx5* conditional alleles were constructed via gene targeting by flanking exons 2 and 3, exon 3, or exon 3, of each gene with LoxP sequences, respectively (Figure S12). Conditional mice were established from each conditional ESC line. *Cbx1*, *Cbx3*, or *Cbx5* mutant mice were next crossed with mice bearing *CreERT2* alleles, to enable excision of the floxed alleles by addition of 4-OH tamoxifen (4-OHT). Triple conditional mutant mice were obtained via crossing the single, or double conditional mice. Triple conditional ESC lines were made in house from the triple conditional mouse blastocysts. ESCs were cultured in D-MEM (Kohjin-bio, #16003550) with 20% fetal bovine serum (Sigma-Aldrich, #172012), MEM nonessential amino acids (GIBCO, #11140-050), L-glutamine (GIBCO, #25030-081), 2-mercaptoethanol (Sigma-Aldrich, #M1753), and LIF (in-house preparation) on mitomycin C-treated (Sigma-Aldrich, #M4287) primary MEF feeder layers. For conditional KO, 4-OH Tamoxifen (SIGMA, #H7904; (Z)−4-Hydroxytamoxifen, ≥98% Z isomer, dissolved in ethanol) was added to medium to a final concentration of 800 nM and cultured for 6 days. The stock solution was prepared as 2 mM (X2500). After 7 days of tamoxifen (4-OHT) induced deletion of HP1 proteins in the HP1cTKO cell line, the cells failed to thrive and cell division became extremely slow. We suspect that changes to 5hmC and 5mC in HP1cTKO ESCs at 6 days are in an intermediate stage, similar to that seen previously in SETDB1 -/- ESCs which were cultured for 4 days[82].

**HEK293T.** The HEK293T cell line was obtained from DSMZ and maintained in DMEM (Life Technologies) supplemented with 10% FBS (Gibco), 1% Penicillin/Streptomycin and 1% GlutaMAX (Gibco). Cell lines were tested routinely for mycoplasma prior to experiments.

HEK293T cells were cultured in DMEM+Glutamax (Gibco) supplemented with 10% FBS and 1% penicillin/streptomycin (Gibco). For Co-IP experiments, HEK293T cells were seeded in 6 well plates at a density of $0.3 \times 106$ cells per well. For IHC, cells were plated onto glass coverslips in 24 well plates at 100,000 cells per well. Plasmid DNA was introduced into cells by chemical transfection using Lipofectamine 2000 according to the manufacturer's instructions.

### Co-immunoprecipitation (Co-IP) and western blot (WB)

Six well plates were rinsed briefly in cold PBS, then lysed on ice in 300 μl of Radioimmunoprecipitation assay buffer (RIPA, 50 mM Tris,

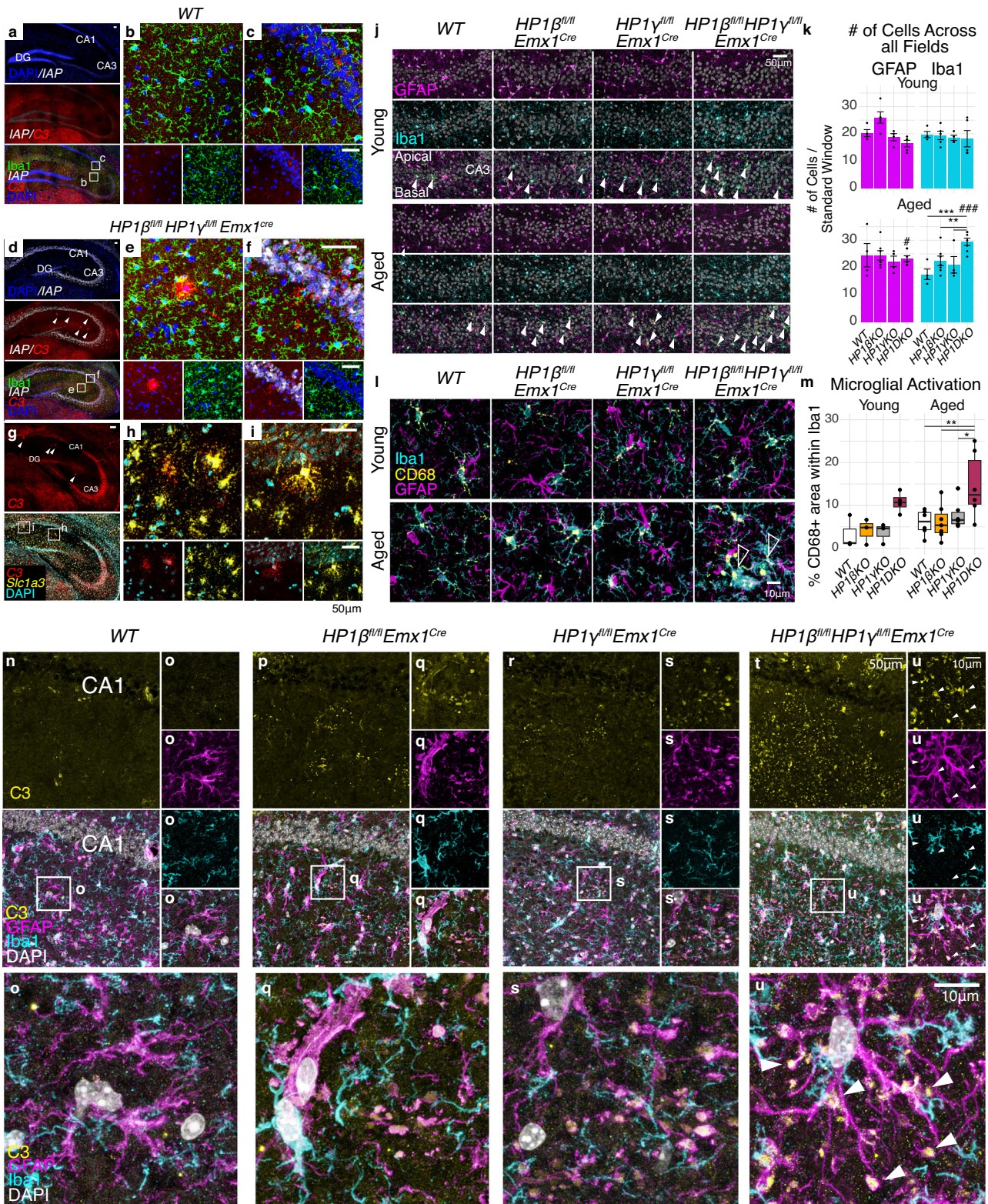

pH 8.0, 150 mM NaCl, 1% Triton X-100, 0.5% Sodium Deoxycholate, 0.1% SDS) supplemented with 1X Protease Inhibitor Cocktail (PIC, Roche). Lysates were then sonicated for 15 pulses on ice using a probe sonicator. Insoluble debris was then precipitated by centrifugation at 13,200 g at 4 °C, and then decanted into a new tube.

Protein concentration was measured using a standard Bicinchroninic acid (BCA) assay, by using 10 µl of protein sample diluted 1:10,

plating 25 µl into a 96 well plate in triplicate along with a Bovine Serum Albumin (BSA) standard (0, 0.1, 0.2, 0.5, 0.75, 1.0, 1.5, 2.0 µg/µl).

Prior to immunoprecipitation, each sample was adjusted to a 300 µl volume at a protein concentration of 1.5 µg/µl using lysis buffer, and 20 µl of lysate was set aside for input. To perform the immunoprecipitation, 1.5 µl of antibody (mouse anti-myc 9B11, Cell Signaling or goat anti-GFP, Rockland) was incubated per sample on a rocker for

**Fig. 5 | Chronic de-repression of ERVs coincides with the appearance of C3+ reactive Astrocytes and increased CD68+ Microglia.** Moderate Complement 3 (C3) RNA can be detected by RNAscope in aged WT (**a**–**c**), where it can be detected in most cells of the hippocampus including CA1 and CA3 pyramids. (Representative images are displayed here after testing across 3 brains per condition). De-repression of IAP transcripts in aged *HP1β^fl/fl^HP1γ^fl/fl^Emx1^Cre^* (white, **d**) corresponds with large C3+ islands (arrows in **d**) that can be found in the *Stratum Radiatum* (magnified in **e**) and emanating from the *Stratum Pyramidale* (magnified in F). Such C3 foci are surrounded by microglia with distinct morphology (compare **b**, **c**–**e**, **f**). C3+ foci found in the *Stratum Radiatum* of *HP1β^fl/fl^HP1γ^fl/fl^Emx1^Cre^* hippocampi are Slc1a3+ astrocytes (**g**, magnified in **h**, **i**). Representative RNAscope images (**a-i**) are from two attempted assays with a third shown in Figure S7. Scalebars (**a**–**i**) are all 50μm. **j** Representative images of CA3 pyramidal layers stained for GFAP, Iba1 and Dapi. Iba1+ cells entering the *stratum pyramidale* are indicated with solid arrows. **k** Quantification of GFAP+ and Iba1+ cells (mean count ± SEM & raw data points) quantified across apical, somal, and basal regions in CA1 and CA3 fields (see also Figure S6). Statistics two-way ANOVA with estimated marginal means post-hoc test with Tukey's familywise correction. $n_{WT(young)} = 8$, $n_{WT(aged)} = 8$, $n_{HP1βKO(young)} = 10$, $n_{HP1βKO(aged)} = 14$, $n_{HP1γKO(young)} = 8$, $n_{HP1γKO (aged)} = 8$, $n_{HP1β/γ DKO (young)} = 10$, $n_{HP1β/γ DKO (young)} = 14$, where n is a single image taken from one of three biological replicates. Adjusted *P* values for genotype test within age (Iba1): aged HP1β/γDKO vs WT

$p = 0.003$, aged HP1β/γDKO vs aged HP1γKO $p = 0.0152$, aged HP1β/γDKO vs aged HP1βKO $p = 0.0152$. Adjusted *P* values for age test within genotype: Iba1 HP1β/γDKO age $p < 0.0001$, GFAP HP1β/γDKO age $p = 0.0087$. **l** Iba1+ microglia that can be found in the *stratum radiatum* surrounding GFAP+ astrocytes contain large CD68+ compartments suggesting endosomal activity. **m** Quantification of Microglial activation (from **l**) measured by the proportion of CD68+ area within Iba1+ microglia. Statistics two-way ANOVA with estimated marginal means post-hoc test with Tukey's familywise correction. $n_{WT(young)} = 3$, $n_{WT(aged)} = 6$, $n_{HP1βKO(young)} = 3$, $n_{HP1βKO(aged)} = 7$, $n_{HP1γKO(young)} = 3$, $n_{HP1γKO (aged)} = 6$, $n_{HP1β/γ DKO (young)} = 4$, $n_{HP1β/γ DKO (young)} = 6$, where n is a single image taken from one of three biological replicates. Adjusted *P* values for genotype test within age: aged HP1β/γDKO vs WT $p = 0.008$, aged HP1β/γDKO vs aged HP1γKO p = 0.0444, aged HP1β/γDKO vs aged HP1βKO p = 0.0064. Box and whisker plots display median, bounds of box at 25th and 75th percentiles, and whiskers to farthest datapoint within 1.5 * the interquartile range. Immunohistochemistry of Complement C3 in aged hippocampi in WT (**n**), HP1βKO (**p**), HP1γKO (**r**), HP1β/γDKO (**t**), with magnifications (**o**, **q**, **s**, **u**, respectively). HP1β/γDKO hippocampi show C3 protein accumulation in the end feet of astrocytic processes (**u**, white arrows). C3 protein accumulation in end feet was observed twice in two separate assays. All raw data and exact *P* values can be seen in the Source Data file.

---

2 hours at 4 °C. During this time, protein G sepharose beads (GE Healthcare, 25 μl per sample) were rinsed 3 × 15 min in 1 ml of cold TBS (50 mM Tris pH 7.5, 150 mM NaCl), rocking at 4 °C. Between washes beads were spun down using a short 5 sec spin on a tabletop centrifuge (no greater than 9000 g). Following the 2 h antibody/lysate incubation, washed beads are added to each sample and incubated on a rocker for an additional hour at 4 °C. Following IP, beads are washed twice using lysis buffer and twice with TBS. On the final wash, as much buffer was removed as possible before addition of 25 μl 2.5X lammeli buffer. IP was then boiled at 95 °C for 5 min. For input samples, 5 μg of total lysate was used in a volume of 25 μl, adjusted with 5 μl 5X lammeli buffer and the appropriate amount of lysis buffer, and boiled at 95 °C for 5 min. A full list of antibodies used in westernblot and ChIP can be found in Supplementary Data 1.

## RNA isolation

For cDNA library construction, RNA was purified from P0 cerebral cortices using TRIzol (Invitrogen) according to the manufacturer's instructions followed by reverse transcription by SuperScript II (Thermo Fischer) using random primers. For RNAseq, RNA was isolated using the Relia-prep RNA mini kit (Promega) from hippocampi of male mice aged 3 months (young timepoint) or 12–13 months (aged timepoint) according to the manufacturer's instructions.

## RNAseq library preparation

RNA-seq libraries were prepared with the NEBNext Ultra RNA Library Prep Kit for Illumina (New England Biolabs), using 1 μg total RNA per experiment.

## Molecular cloning

For in situ probes, primers were designed around consensus sequences obtained from repbase (https://www.girinst.org/repbase/) and amplicons were queried using UCSC's BLAT and in silico PCR tools to determine the estimated diversity of transcripts corresponding to the probe. Probe primers for a unique IAP element on chr 2 were inferred based on unpublished qPCR primers (Julie Brind'Amour, UBC) (Supplementary Data 1). Probe sequences were amplified from cDNA using GoTaq Polymerase (Promega) and ligated into the pGEM®-T vector (Promega). Linearized plasmids were then used as templates for in vitro transcription using either SP6 or T7 (Roche) using DIG labeled nucleotides (Roche). Following DNA digestion and RNA purification, the RNA probe was resuspended in 20 μl water and 180 μl hybmix (50% formamide, 5X SSC pH 7.0, 1% Boehringer block, 5 mM EDTA, 0.1%

Tween-20, 0.1% CHAPS, 0.1 mg/ml Heparin, 100 μg/ml Yeast tRNA) and stored at −20 °C until needed.

To clone HP1 and SUV420H2 expression constructs, two rounds of PCR were conducted using the high fidelity Q5 Polymerase (New England Biolabs). The first round of PCR amplified the 'naked' coding region of the gene (*_110 and *_111 primers), and using this product as a template, a second PCR was performed with primers removing the stop codon and containing restriction sites. These amplicons were then A-tailed using GoTaq polymerase (Promega) and ligated into the pGEM®-T vector (Promega) yielding pGEMT-KnpI-HP1x (where x is α, β, or γ). To generate eGFP fusions, inserts were digested with KpnI and AgeI for insertion into pCAG-eGFP (Clontech), yielding pCAG-HP1x-GFP fusion constructs. Prior to cloning SUV420H2, pCAG-mycDKK was created by substituting the CAG promoter from pCAG-eGFP with the CMV promoter of pCMV6-Entry though digestion with SpeI and EcoRI. First, 'naked' SUV420H2 was amplified using 110 and 111 primers, then 1 μl of this template was used for a PCR using kpnI and XhoI primers. The KpnI-SUV420H2-XhoI PCR product was then gel-extracted and digested for 20 min using Fast Digest KpnI and XhoI (Fermentas) and subsequently ligated into KnpI/XhoI digested pCAG-mycDKK, resulting in an in-frame C-terminal myc-DKK (Myc tag and Flag tag).

For SUV420H2/HP1 IP experiments, mutations were created in the chromoshadow domain of HP1γ in pGEMT-KpnI-HP1γ using the Q5 site-directed mutagenesis kit (New England Biolabs) according to manufacturer's instructions (Supplementary Data 1). Following sequence verification, mutant HP1γ inserts were ligated into pCAG-eGFP as above. Expression plasmids from this study have been deposited at https://www.addgene.org/browse/article/28223393/.

For cloning of the full length IAP construct for in vitro transcription, pFL, a plasmid encoding full length IAP[83] and kind gift of Prof. Horie, was first subcloned to delete the antisense intronic GFP using a NEB KLD reaction, and then a T7 promoter sequence was added 5' to the LTR by NEBuilder gateway cloning, resulting in the plasmid pT7-IAP. Primers can be seen in Supplementary Data 1.

## In vitro transcription of IAP ssRNA

The pT7-IAP plasmid was linearized using NaeI and NotI for 4 h at 37 °C using 15ug of plasmid DNA. The linearized 6.8 kb fragment was then gel purified & used as a template for in vitro transcription (IVT) using the RiboMax large scale RNA production system (Promega, cat # P1300) with minor modifications. First, 3' overhangs were filled in by incubating 10ug of DNA template with DNA polymerase I large (Klenow) Fragment in T7 transcription buffer for 15 min at 22 °C. Then T7

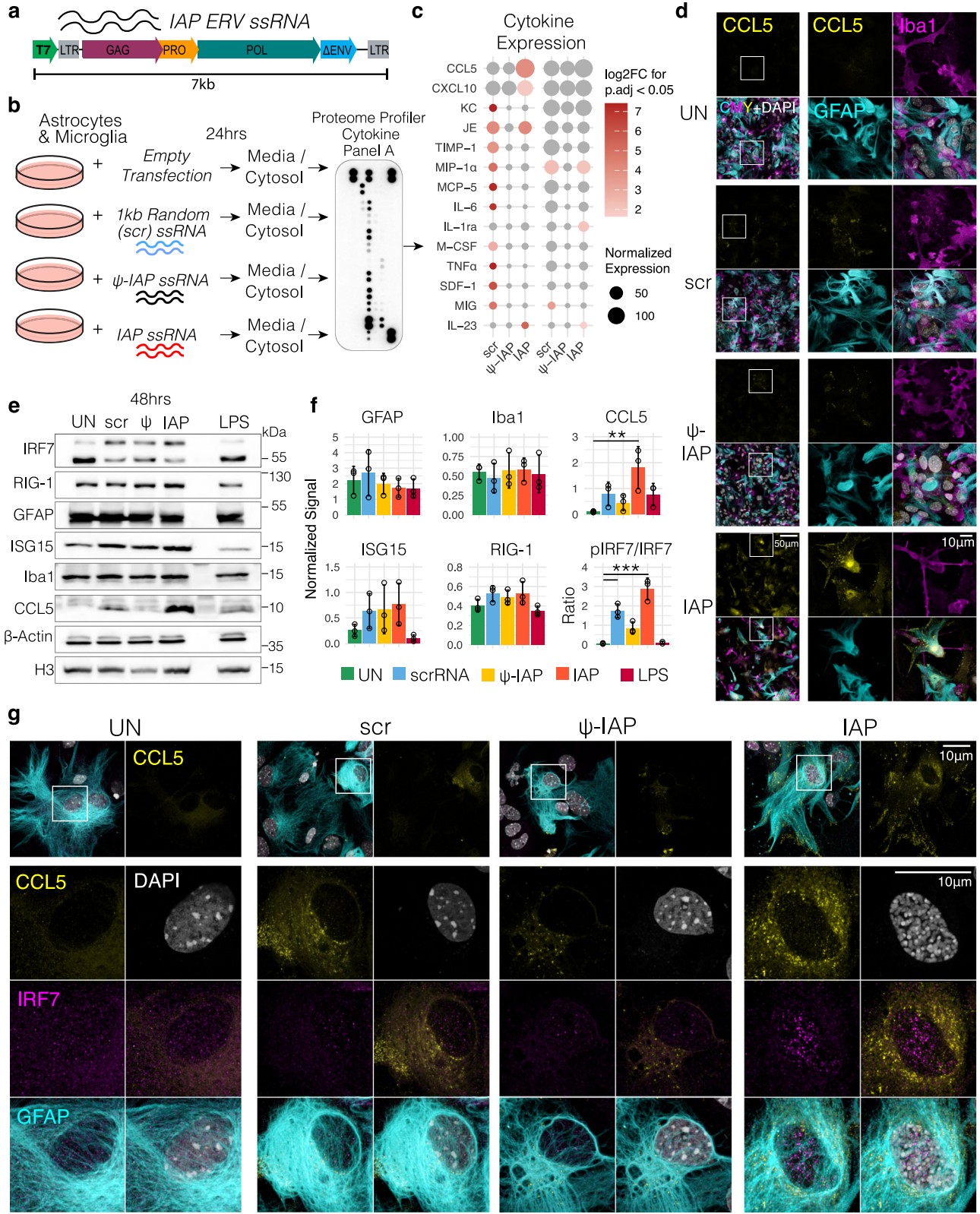

in vitro transcriptions were set up with rNTP mixes. For IAP RNA a normal mix of rNTPs (100 mM ATP, 100 mM UTP, 100 mM CTP, 100 mM GTP), and for pseudo-IAP, pseudo-uracil was substituted (100 mM ATP, 100 mM Ψ-UTP, 100 mM CTP, 100 mM GTP). IVT reactions were then incubated with T7 & respective rNTP mixes for 4hrs at 37 °C. Following in vitro transcription, RQ1 RNase-free DNase was added at a concentration of 1 u per μg of template DNA and incubated

for 15 mins at 37 °C. Then, RNA was extracted in the aqueous phase after addition of 1 volume of citrate-saturated phenol (pH 4.7):chloroform:isoamyl alcohol (125:24:1) (Sigma, cat # 77619), and a second addition of 1 volume of chloroform:isoamyl alcohol (24:1). The aqueous phase was then transferred NAP-5 (GE Healthcare Cat# 17-0853-01) chromatography columns for the removal of unincorporated ribonucleotides and eluted using ultrapure RNase free water. RNA was

**Fig. 6 | Acute introduction of IAP ssRNA induces an inflammatory response in astrocytes. a** Domain map of full length IAP sequence used for generation of ssRNA. **b** Schematic showing experimental design where primary astrocytes and microglia mixed cultures are exposed to a pulse of transfection reagent (untreated, UN), a 1 kb random scrambled ssRNA (scr), IAP-ssRNA, or pseudo (ψ)- IAP ssRNA, where ψ-IAP is generated from the same template but with ψ-uracil. Media and cytosolic lysates were then incubated with Proteome Cytokine Array Panel A. **c** Summary quantification of significant changes to cytokine expression following acute ssRNA incubation. Cytokine changes are plotted as dots proportional to their normalized expression. Statistically significant changes (one-way ANOVA for condition by gene followed by Dunnett's, adjusted *P* value < 0.05) are colored according to their log2Fold change over untreated. $UN_{Media} n = 4$, $UN_{Cytosol} n = 3$, $scr_{Media} n = 3$, $scr_{Cytosol} n = 3$, $\psi\text{-}IAP_{Media} n = 3$, $\psi\text{-}IAP_{Cytosol} n = 3$, $IAP_{Media} n = 3$, $IAP_{Cytosol} n = 3$, where *n* is a biological replicate. **d** Immunofluorescence stain of mixed glial cultures comprised of Iba1+ microglia and GFAP+ astrocytes stained for CCL5 (RANTES), repeated once for each of the corresponding 3 biological

replicates in (**e**). **e** Western blot of whole cell lysates from mixed cultures measured 48 hrs after being treated with transfection reagent only (untreated, UN), random scramble ssRNA (scr), ψ-IAP ssRNA (ψ), IAP-ssRNA or Lipopolysaccharide (LPS). **f** Quantification of western blots (mean ± SD) where lower molecular weight proteins (CCL5, ISG15, Iba1) were normalized to H3 and remaining higher molecular weight proteins normalized to β-actin over *n* = 3 biological replicates. Box and whisker plots display median, bounds of box at 25th and 75th percentiles, and whiskers to farthest datapoint within 1.5 * the interquartile range. One-way ANOVA followed by Dunnett's test, $CCL5_{IAP\ vs\ UN}\ p = 0.00441283$, $pIRF7/IRF7_{IAP\ vs\ UN}$ $p = 0.0000022503$, $pIRF7/IRF7_{scrRNA\ vs\ UN}\ p = 0.00078752$. **g** Airyscan super-resolution images of mixed glial cultures following 48hr incubation with IAP and control ssRNAs. Top row with IRF7 in magenta, GFAP cyan, CCL5 in yellow and DAPI in white and single CCL5 channel images. Magnification in the bottom panel shows single channel images, with the bottom row displaying merged CMY images and CMY + DAPI images. CCL5 and IRF7 accumulation was observed consistently in two separate assays. All raw data and exact *P* values can be seen in the Source Data file.

then precipitated by addition of 0.1 volume of 3 M Sodium Acetate (pH 5.2) and 1 volume of isopropyl alcohol, mixing well and incubating 5 min on ice. Precipiated RNA was then pelleted by centrifugation at 13,000 in a tabletop centrifuge at 4 °C. Supernatant was then discarded and the RNA pellet washed in 70% ultrapure EtOH, dried & resuspended in a volume of Ultrapure ddH2O equal to the transcription reaction. RNA concentration was then measured by first diluting 2 µl of RNA into 298 µl of water and measuring the absorbance at 260 nm. The concentration of RNA was then calculated using the expression

$$C = (A_{260nm} * dilution\ factor)/(10313\ x\ nucleotides)$$

where *C* is in moles and the dilution factor is 100. In vitro transcribed RNA was then confirmed by denaturing 1 µg of RNA at 65 °C for 10 min in 1.5X probe buffer (60% formamide, 40% glycerin 6% formaldehyde, 5% ethidium bromide, 5% bromophenol blue, 20 mM MOPS, 5 mM EDTA, 2.1 mM Calcium Acetate) and running on a 1% agarose gel (20 mM MOPS, 5 mM EDTA, 2.1 mM Calcium Acetate, 6% formaldehyde, 1 % agarose) using MOPS (20 mM MOPS, 5 mM EDTA, 2.1 mM Calcium Acetate) as the running buffer.

### Generation of random ssRNA
A random 1000 bp sequence was generated in python using

```
import random
def generate_random_dna(length):
  return ''.join(random.choice('ATCG') for _ in range(length))
random_dna = generate_random_dna(1000)
```

This random sequence (exact sequence in Supplementary Data 1) was synthesized by BioCat and inserted into a pBluescript II SK (-) backbone containing a T7 promoter. Following plasmid linearization, Random ssRNA was then transcribed using the RiboMax large scale RNA production system (Promega, cat # P1300), as described above.

### Transfection of Primary Astrocytes & Microglia for Cytokine Profiling
Astrocytes and microglia were obtained by trypsin-assisted dissociation of P0 cortices and cultured for two weeks in DMEM supplemented with 10% FBS and 1% penicillin/streptomycin. Mixed glia were then seeded on poly-D-lysine coated plates at 500,000 cells per cell of a six well plate and 100,000 cells per well of a 24 well plate. For one biological replicate 4 wells of a six well plate per pooled per experiment. RNA was put into complex with lyovec (Invivogen, cat # lyec-1) to a final concentration of 10 ng/µl, and 100 µl (1 µg RNA) applied per six well and 25 µl (250 ng RNA) applied per 24 well. After a twenty-four hour incubation, conditioned media was removed and cells were

rinsed in cold PBS before being lysed in 300 µl (across 4 wells of a six well plate) using cytokine lysis buffer (1% Igepal CA-630, 20 mM Tris-HCl (pH 8.0), 137 mM NaCl, 10% glycerol, 2 mM EDTA, 10 µg/mL Aprotinin, 10 µg/mL Leupeptin, and 10 µg/mL Pepstatin). Cells were lysed for 30 mins on ice with gentle pipetting. Proteome Profiler Cytokine Array Panel A (R&D Systems, cat # ARY006) was then used for parallel detection of activated cytokines and chemokines in cytosolic lysates or conditioned media according to manufacturer's instructions. In the 48 hr paradigm, 100 ng/ml LPS was included as a positive control, and cultures were lysed directly in 2X lammeli buffer.

### Analysis of cytokine panel
Quantification of dot plots was performed using the Quick Spots software (Ideal Eyes Systems, Inc.). Technical replicates were averaged and then percent normalized within membrane by dividing the gene value by the membrane reference spot value. Log2 fold change was then calculated by log2 of the ratio of the mean gene value divided by the control value for that gene in that fraction (media / cytosol). Analysis of variance was performed on normalized values within fraction and genes with significant changes were tested against lyovec-only control values using Dunnett's post hoc test. Full tables of values and statistics can be found in the source data.

### In situ hybridization
Standard DIG-labelled in situ hybridization was performed using the method described in ref. [84]. Multiplex in situ hybridization was performed using RNAscope (Advanced Cell Diagnostics) according to the manufacturer's instructions using a custom designed probe for IAP and standard C3 and Slc1a3 probes. Iba1+ cells were identified after RNAscope using an Iba1 antibody (Wako).

### In utero electroporation
In utero electroporation was performed according to the initial protocol[85] with minor modifications. All surgical procedures were performed in accordance with LaGeSo experimental licenses G0079/11 and G0206/16.

### Tissue processing & histology
For embryonic tissue, the date of the vaginal plug was counted as embryonic day 0.5 (E0.5). Pregnant females at the desired stage were killed by an i.p. injection of 600 mg pentobarbital per kg body weight and death was confirmed by cervical dislocation. Following a midline incision, uterine horns were excised and placed in a dish containing ice cold PBS, where embryonic brains were isolated using microforceps. Brains were immediately fixed in paraformaldehyde (PFA, 4% in PBS) overnight at 4 °C. For animals older than P6, animals were given a lethal i.p. injection of 600 mg pentobarbital per kg body weight and transcardially perfused with PBS (10-20 ml, depending on age) until the liver

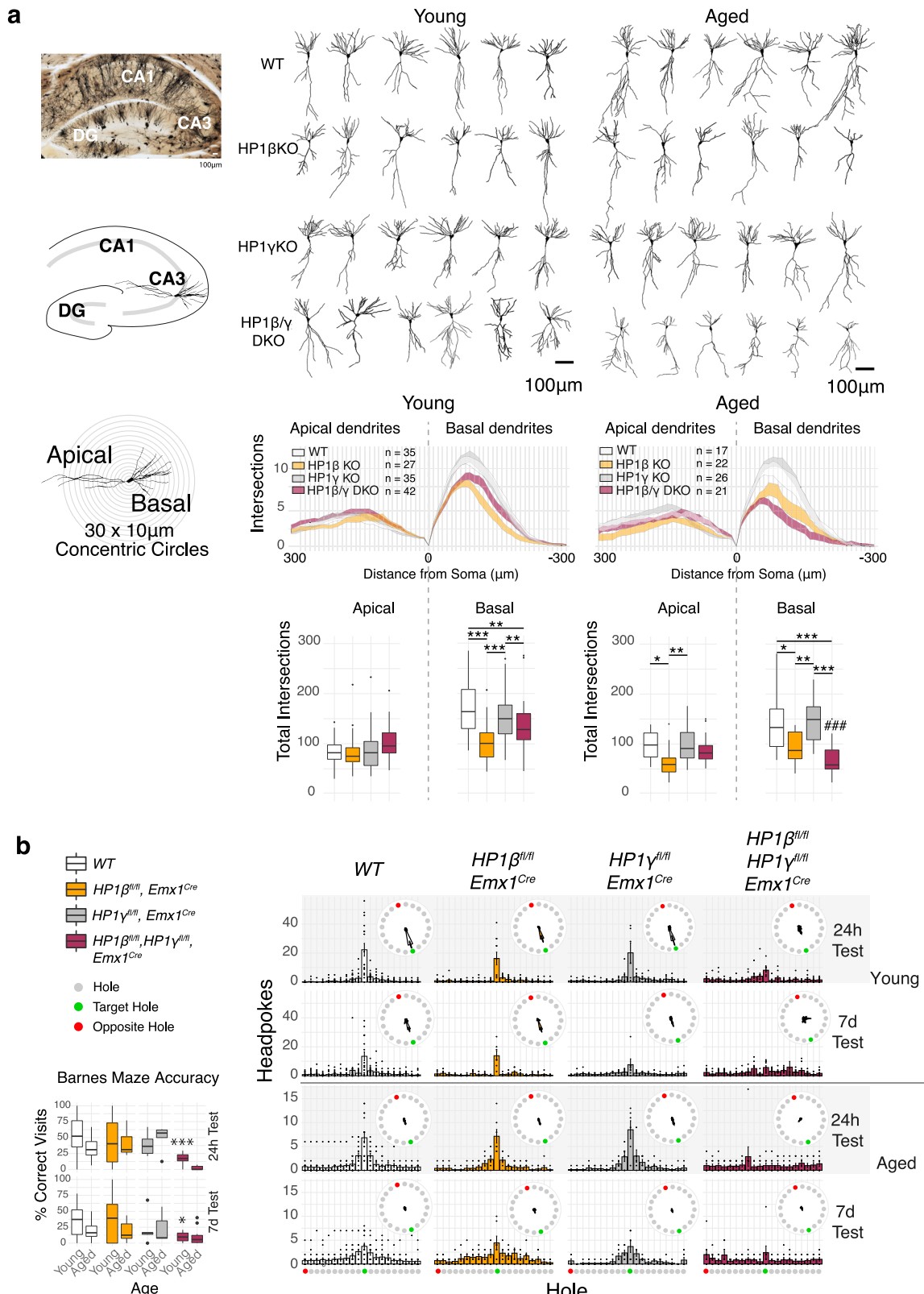

**a**

**b**

was clear, followed by perfusion with 4% PFA (5–20 ml, depending on age). Brains were then isolated and fixed overnight in 4% PFA.

For in situ hybridization, all solutions were composed using ddH$_2$O treated with Diethyl pyrocarbonate (DEPC, Sigma, prepared by shaking 1 ml DEPC in 1 L ddH$_2$O at 37 °C O/N) and per processed by cryosectioning. For cryosectioning, after fixation brains were dehydrated in sequential concentrations of sucrose (15%, 30% in PBS)

before being embedded in Optimal cutting temperature compound (OCT, Tissue-Tek) and freezing on dry ice. Tissue blocks were stored at −20 °C until 16 μm sectioning on a cryostat at −20 °C, when they were collected on positively charged slides (Superfrost, ThermoFischer) and allowed to dry for 1 hr before being re-frozen for storage at −20 °C.

For paraffin sectioning, fixed brains were dehydrated by ethanol row (30% EtOH, 50% EtOH, 70% EtOH, 80% EtOH, 90% EtOH, 100%

**Fig. 7 | HP1 deficiency causes age related behavioral abnormalities and CA3 dendritic tree degeneration. a** CA3 Dendritic Complexity in young and aged HP1 mutants by Golgi impregnation and Scholl analysis. Deficits in basal dendrite complexity can be observed in young HP1βKO and HP1β/γ DKO animals. While basal dendrite complexity is nearly identical in aged HP1βKO, HP1β/γ DKO basal dendrites show an age dependent degeneration, losing almost 50% of their complexity. $n_{WT(young)} = 35$, $n_{WT(aged)} = 17$, $n_{HP1βKO(young)} = 27$, $n_{HP1βKO(aged)} = 14$, $n_{HP1γKO(young)} = 35$, $n_{HP1γKO (aged)} = 23$, $n_{HP1β/γ DKO (young)} = 42$, $n_{HP1β/γ DKO (young)} = 21$, where n is # of neurons counted over 3 biological replicates. Two-way ANOVA with Bonferroni adjustment on multiple comparison (two tailed). **b** Histograms of performance in the circular Barnes Maze (mean count ± SEM & raw data points, green = target hole, red = opposite). Histograms plot mean headpokes ±SEM. of $n_{WT(young)} = 18$, $n_{WT(aged)} = 16$, $n_{HP1βKO(young)} = 11$, $n_{HP1 β KO(aged)} = 7$, $n_{HP1γKO(young)} = 6$, $n_{HP1γKO (aged)} = 5$, $n_{HP1β/γ DKO (young)} = 11$, $n_{HP1β/γ DKO (young)} = 12$. Two way ANOVA, Bonferroni adjustment on multiple comparison, two-tailed. HP1β/γ DKO$_{young}$ vs WT$_{young}$ 24 h test $p = 0.0009$. HP1β/γ DKO$_{young}$ vs WT$_{young}$ 24 h test $p = 0.0081$. Box and whisker plots (**a**, **b**) display median, bounds of box at 25th and 75th percentiles, and whiskers to farthest datapoint within 1.5 * the interquartile range. All raw data can be seen in the Source Data file.

EtOH) followed two changes of Xylol and two changes of paraplast before casting in paraffin in metal embedding molds. For sectioning, a Microtome (Leica) was used and 14 μm thick sections were collected in 37 °C ddH$_2$O on positively charged slides (Superfrost, ThermoFischer).

Nissl Stains were performed by incubation with cresyl violet. Cresyl violet staining solution was prepared by dissolving 0.1 g cresyl violet acetate in 100 ml ddH$_2$O O/N. Following addition of 10 drops (-0.3 ml) glacial acetic acid, this cresyl acetate solution was filtered. Nissl stain was performed on rehydrated paraffin sections (Xylol II: 5 mins, Xylol I: 5 mins, 3 × 5 min 100% Ethanol, 3 min 95% Ethanol). Sections were immediately stained in 0.1% Cresyl violet for 3–10 min, followed by rinsing in dH$_2$O to remove excess stain. Sections were then differentiated in 95% ethanol for 2–30 min, checking microscopically for optimal staining. Sections were then dehydrated by 2 × 5 min 100% ethanol washes and cleared with xylol (2 × 5 mins) before mounting with Entellan (Sigma).

Immuno Histochemistry (IHC) was performed primarily on cryosections. For paraffin sections, prior to IHC sections were rehydrated and an antigen retrieval step (Boiling 3 × 5 mins in Antigen Unmasking Solution, Vector Labs) was performed prior to blocking. To perform IHC, slides were washed 2 × 5 min in PBS, then blocked and permeabilized in blocking solution (2% BSA, 1% Triton X100 in PBS). All further antibody steps use this same blocking solution as diluent. Primary Antibodies were diluted 1:200–500 in blocking solution and incubated on sections at 4 °C O/N. The following day, slides were washed 3 × 10 min in PBS and then incubated for two hours at room temperature with the appropriate secondary antibody (Dianova). Sections were then washed 2 × 5 mins and stained with Hoechst/DAPI (1:5000 in PBS) for 5 min at room temperature. When using adult sections, lipofuscin autofluorescence was quenched by a 10 min incubation with a solution containing 10 mM CuSO$_4$ & 50 mM NH$_4$Cl. Sections were then mounted aqueously using Immu-Mount (Shandon). A full list of antibodies used can be seen in Supplementary Data 1.

For Golgi impregnation, fresh brain samples from WT, HP1β$^{fl/fl}$Emx1$^{Cre}$, HP1γ$^{fl/fl}$Emx1$^{Cre}$ and HP1β/γ$^{fl/fl}$Emx1$^{Cre}$ of young (3 months) and aged (13–14 months) mice were cut into two hemispheres and impregnated in Golgi-Cox solution for 2 weeks as described in ref. 86. Sholl analysis[87] was performed blind on CA3 hippocampal neurons by using the concentric circles and cell counter plug-ins available for ImageJ. Intersections were quantified across thirty 10 μm spaced concentric circles. The Simple neurite tracer plugin (ImageJ) was used to draw representative neurons.

The number of Prox1+ and Ki67+ cells in dentate gyrii were quantified by creating a pipeline in CellProfiler (http://cellprofiler.org). Images were first masked such that only the dentate gyrus was visible, then split into individual files by RGB channel (DAPI- blue, ki-67 – green, Prox1-red). Primary objects were identified for each channel using the following parameters: For nuclei, min-diameter 8, max-diameter 14, threshold correction 1.5, distinguishing by shape. For ki67, min-diameter 5, max-diameter 25, threshold correction 1, distinguishing by intensity. Prox1 primary objects were identified using the same settings as nuclei. Primary objects were then related to nuclei, removing false signal, and count was exported to a csv file. To calculate cells/μm, the area of the dentate gyrus measured was quantified manually in Fiji/ImageJ[88] (https://imagej.nih.gov/ij/).

The percentage of area occupied by CD68 inside Iba1+ microglia was quantified by creating a pipeline in CellProfiler and analyzing images in three batches. Images were first corrected for illumination, then aligned, and then Iba1 primary objects were identified with min pixel size 4 max 150 using an adaptive Otsu threshold strategy with three classes, identifying the foreground and a 0.1 lower threshold bound and an adaptive window size of 65. Second and third batches of CD68 stains required minor adjustments to the lower threshold bound for background correction.

## Behavioral experiments

For behavioral experiments, A total of 42 male animals (15 WT, 11 HP1βKO, 6 HP1γKO, 10 HP1β/γDKO) completed the aged time point, while 4 died (2 WT, 1 HP1βKO, 1 HP1β/γDKO) between young and aged testing. Box and whisker plots for behavioral experiments are comprised of median and 25th and 75th percentiles, where whiskers extend no further than 1.5X the interquartile range. All line charts plot the mean, with standard error rendered as a ribbon surrounding the line. Unless stated otherwise, behavioral experiments were analyzed using Two way ANOVA with Bonferroni correction for multiple comparisons; where Asterisks (*) denote tests to between genotype within age (* = $p < 0.05$, ** = $p < 0.01$, *** = $p < 0.001$) and hashtags (#) denote tests within genotype between age (# = $p < 0.05$, ## = $p < 0.01$, ### = $p < 0.001$).

All behavioural experiments were undertaken in the Animal Outcome Core Facility (AOCF) at the Charité. Behavioral tests were performed within the guidelines granted by the LaGeSo under an extension to the experimental license G0079/11 and G0206/16. Prior to behavioral testing, male HP1FEC mice were implanted with subdermal RFID transponders to ensure accurate identification. Behavioral experiments were performed on both young adults (3 months) and aged adults (13-14 months). Prior to each cohort of behavioral testing, all animals were subjected to a modified **S**mithKline Beecham, **H**arwell, **I**mperial College, **R**oyal London Hospital, **p**henotype **a**ssessment (SHIRPA), which ensured animals did not have any gross deficits in vision, audition, grip strength, pina reflex and normal exploratory locomotion. After SHIRPA assessment, behavioral tests were always carried out at the same time of day, with tests spanning a 1 month period. Tests always occurred in the following order: Open Field Activity, Paired-Pulse inhibition, Barnes Maze, Social Activity Monitor, HomeCageScan, Nest Construction.

**Open field activity.** Animals were placed in the center of a square enclosure for 10 minutes while an overhead camera records and movement is tracked using the Biobserve Viewer Software. Activity in the 'center zone' and periphery were binned per minute.

**Paired-pulse inhibition.** Animals were tested two at a time in a 2-box startle box apparatus (TSE systems), which consisted of black soundproofed plexiglass boxes (49 cm × 49 cm × 49 cm). The floors of internal cases were composed of metal bars connected to pressure sensors, which enabled precise measurement of startle response. Upon program start, animals acclimatized for 5 min, followed by a

program (randomized by trial) that tested response to startle pulse alone (120 dB for 40 ms) or response to the pulse that had been preceded by a pre-pulse (one of 69 dB, 73 dB or 81 dB for 20 ms).

**Barnes maze.** The Barnes Maze is a circular platform containing 20 holes around the circumference, one hole contains a submerged nest that serves as an escape from the open environment. Animals were trained on the location of the submerged nest. This involved placing the animal in the center of the platform while loud static noise is played over four 3-min trials over 4 training days, in which time the mouse could often find the submerged nest. If the mouse could not find the nest by the end of the 3 min, the mouse was shown the nest. Once in the nest the mouse was allowed to stay for 30 seconds to allow for positive reinforcement. Following 4 days of training, the nest is removed and animals were tested on the 5th day (24 h test) where mice were placed in the center of the platform and their hole seeking behavior is recorded (time to target hole, errors before target hole, headpokes per hole) over a 90 second testing period. Animals were then tested one week later (7day test) in the same manner.

**Social activity monitor (SAM).** To measure basic interaction and circadian activity, animals in their home cages were placed on top of RF sensors that detect motion. Animal activity was then recorded for 14 days using Phenoscoft control software. SAM data was binned by hour and following data export, RFIDs were decoded to corresponding animal ids and genotypes.

**HomeCageScan (Microbehaviors).** Animals were recorded individually over 24hs using CleverSys Software for any changes in sterotyped murine behavior. Prior to recording, background cage registration, night/day and transition calibrations were set according to each cage. For data export, data was binned by minute and by hour.

**Nest construction.** Because a pilot experiment revealed that a two day separation of male animals from their homecages resulted in hyper aggression upon their return, nest construction was only tested in the aged timepoint. To test nest construction ability, mice were housed individually and given a square piece of densely woven cotton called a 'nestlet' (Ancare). Animals were allowed to habituate with the nestlet for the first 24 hrs. For the second 24hs a new nestlet was supplied and the following morning what remained of the nestlet was weighed and the complexity of the nest was scored based on the standard rubric: (1) The Nestlet is largely untouched (>90% intact). (2) The Nestlet is partially torn up (50–90% remaining intact). (3) The Nestlet is mostly shredded but often there is no identifiable nest site: <50% of the Nestlet remains intact but <90% is within a quarter of the cage floor area, i.e. the cotton is not gathered into a nest but spread around the cage. Note: the material may sometimes be in a broadly defined nest area but the critical definition is that 50–90% has been shredded. (4) An identifiable, but flat nest: >90% of the Nestlet is torn up, the material is gathered into a nest within a quarter of the cage floor area, but the nest is flat, with walls higher than mouse body height (curled up on its side) on less than 50% of its circumference. (5) A (near) perfect nest: >90% of the Nestlet is torn up, the nest is a crater, with walls higher than mouse body height on more than 50% of its circumference.

## Magnetic resonance imaging

MRI was performed at a 7 Tesla rodent scanner (Pharmascan 70 / 16, Bruker, Ettlingen, Germany) with a 20 mm diameter transmit/receive volume resonator (RAPID Biomedical, Rimpar, Germany). For imaging the mouse brain a T2-weighted 2D turbo spin-echo sequence was used (imaging parameters TR/TE = 5505 ms/36 ms, rare factor 8, 6 averages, 46 axial slices with a slice thickness of 0.350 mm, field of view of 2.56 × 2.56 cm, matrix size 256 × 256; scan time 13m12s). MRI data were registered on the Allen mouse brain atlas (ABA) using an in-house

developed MATLAB toolbox ANTx (latest version available under https://github.com/ChariteExpMri/antx2). The volumes of each single ABA brain structure were calculated using the back-transformed atlas which matched the individual T2-weighted images[89]. For section-wise analysis, the mean isocortex volume per section per genotype per age was calculated and tested using two-way ANOVA.

## Site specific 5hmC and 5mC analysis by qPCR

Genomic DNA was purified from frozen hippocampi using NucleoSpin Tissue columns (Macherey-Nagel) according to the manufacturer's instructions. Purified gDNA was subsequently processed using the EpiMark 5hmC Analysis kit (NEB, cat# E3317) according to the manufacturer's instructions. CpGs of enzymatically prepared samples were then profiled using primers designed to amplify over an area containing a single HpaII/MspI site. Primers used for qPCR: Nnat: ACCCCTCCTTCTCAACATCC & CGCCGAGGTCTACTGGTCT. For IAPEz: CTTTGAAGGAGCCGAGGGTG & AAGCCTGTCTAACTGCACCAA. qPCR was performed using the GoTaq qPCR Master mix (Promega) including the CXR reference dye on a StepOnePlus Thermocycler (Applied Biosystems).

## Chromatin immunoprecipitation (ChIP) & library construction

ChIP experiments and subsequent library preparation, were performed as previously described[90]. Antibodies used for ChIP can be seen in Supplementary Data 1.

## Reduced representation bisulfite sequencing (RRBS)

RRBS on HP1cTKO cells was carried out following a previously described method[91] with minor modifications. 500 ng of genomic DNA was used as a starting material. Bisulfite conversion was done by the EZ DNA Methylation Gold Kit (Zymo Research, #D5005) with 50 ng of DNA, per sample. 2x KAPA HiFi Hot Start Uracil+ Ready Mix (KAPA Biosystems, #KK2801) was utilized for library amplification. PCR amplification was done for 10 cycles.

## 5mC & 5hmC Joint profiling from hippocampal lysates

Hippocampi were isolated in ice cold PBS and DNA was using the QIAamp Fast DNA Tissue Kit (QIAGEN, cat # 51404). Purified DNA was then used to generate bisulfite converted or oxidative bisulfite converted RRBS libraries using the Ovation RRBS kit (TECAN, formerly NuGEN, cat # 0553-32). RRBS libraries were sequenced in a 50 bp paired end configuration with an additional 6 bp allotted to the library index on an Illumina Novoseq 6000.

## Bioinformatics & data processing

Bioinformatic pipelines were written using Snakemake https://snakemake.readthedocs.io/en/stable/ and deployed on the cluster hosted by the Berlin Institute of Health (BIH). Scripts for analysis are provided at https://github.com/qoldt/HP1-Deficiency-Neurodegeneration.

## RNAseq

For RNAseq data, fastq files were aligned to GRCm38.p5 using STAR[92] with the following settings to maximize repeat mapping (--outFilterMultimapNmax 100, --winAchnorMultimapNmax 100, --outSAMstrandField intronMotif). The TETranscripts[93] package was used to generate a count table using gencode.vM16.basic.annotation.gtf and the prepared repeat masker file. The outputted count table was re-annotated using biomaRt[94] and analyzed for differential expression using edgeR[95]. Hierarchical clustering and heatmap of significantly changed transcripts determined from testing between all WT and all HP1DKO (689 genes, adjusted $p < 0.05$, Supplementary Data 1) was created from a scaled matrix using heatmap2.

No Differentially expressed genes could be detected between young HP1γKO and aged HP1γKO. Young HP1βKO and aged HP1βKO showed 184 differentially expressed transcripts among which were

increases in C4 and C1qa in aged HP1βKO. A non-negligible batch effect meant direct comparison of young HP1DKO and aged HP1DKO was not statistically advisable.

For IAP RNAseq coverage profiles, known IAP coordinates were obtained from the UCSC table browser in bed format. Deeptools[96] was used to generate RPKM normalized 50 bp bin bigwig files with the option --extendReads from aligned RNAseq data. Comatrices were computed to scale regions to an internal size of 500 bp with before and after region lengths of 1000 bp. Read coverage was plotted over IAP elements for each genotype at each age using RPKM (per bin) = number of reads per bin / (number of mapped reads (in millions)* bin length (kb)). Chimeric transcripts were detected using a combination of LIONS[97] and use of a 1000 bp running window filter in Seqmonk. Inflammatory response was profiled by cross referencing genes differentially expressed in HP1DKO with the interferome database[98]. Gene set enrichment and leading edge analysis was performed using GSEA to query raw count data from aged HP1DKO and wildtype against c2.cp.reactome.v6.2.symbols.gmt using Signal2noise in gene_set mode with default parameters. GSEA output was imported into cytoscape using the EnrichmentMap[99] plugin (Jaccard Overlap combined cut-off = 0.375, k constant = 0.5, node cut-off Q = 0.6 and edge cutoff similarity of 0.53). Network node clusters were coarsely annotated using the AutoAnnotate[100] plugin which was further refined using Adobe Illustrator.

Locus specific TE analysis was performed using the locus specific SalmonTE* method described by Schwarz et al.[101]. Briefly, this involved generating a salmon index from repeat masker (mm10) using the alignToFasta.sh helper script provided. Reads were then quantified against this index using SalmonTE quant. Merged counts were then tested for differential expression using DESeq2.

**RRBS.** 5mC and 5hmC RRBS data derived from HP1FEC hippocampi paired bisulfite (BS) and oxidative bisulfite (oxBS) reactions was analysed as follows: Reads (R1 and R3 in this case) were trimmed using trim_galore (https://github.com/FelixKrueger/TrimGalore) with the parameters --paired -a AGATCGGAAGAGC -a2 AAATCAAAAAAAC. Reads were then further processed using the NuGEN diversity trimming script (https://github.com/nugentechnologies/NuMetRRBS/blob/master/trimRRBSdiversityAdaptCustomers.py) and aligned to the GRCm38 (mm10) Bisulfite Genome using Bismark with -p 2 -N 1 --multicore 8. Coverage files were generated using bismark_methylation_extractor with -p --ignore_r2 3. True 5mC was taken directly from the oxBS data. To infer 5hmC state, oxBS coverage files were subtracted from BS coverage files outlined in the created 'Extract 5hmC.R' script, where count $Count_{5hmC} = Count_{BSmethylated} - Count_{oxBSmethylated}$, $\%_{5hmC} = \%_{BSmethylated} - \%_{oxBSmethylated}$, and the number of unmodified cytosines $Count_{NoShmC}$ is calculated based on the relationship $\%_{5hmC} = \frac{Count_{5hmC}}{Count_{5hmC} + Count_{NoShmC}} * 100\%$, rounded to the closest integer. In fringe cases where oxBS signal is higher than BS (5hmC is negative), 5hmC is set to zero and $Count_{NoShmC}$ is set to $Count_{BSunmethylated}$. Coverage files from 5mC and 5hmC were tested for differential methylation using methylkit[102] and intersected with genomic annotations obtained from the UCSC table browser. Odds ratios for each annotation tested were calculated as follows; given $D_{overlaps}$, the number of Differentially Methylated Regions (DMRs) that overlap with the annotation, $N_{overlaps}$, the number of Non-Differentially Methylated Regions (Non-DMRs) that overlap with the annotation, $D$, the total number of DMRs and $C$, the total number of observed cytosines:

$$Odds\ Ratio = \frac{\frac{D_{overlaps}}{D - D_{overlaps}}}{\frac{N_{overlaps}}{C - D - N_{overlaps}}}$$

The probability $P$ of drawing $D_{overlaps}$ or more by chance, when drawing $D$ DMRs from a population of $C$ total cytosines, of which $D_{overlaps}$ and $N_{overlaps}$ are successes (i.e., overlaping with the annotation) was calculated using the hypergeometric distribution. This calculation was performed with the 'phyper' function in R which tests the null hypothesis that the observed number of overlaps is as expected by chance: $P(X \geq D_{overlaps}) = 1 - phyper(D_{overlaps} - 1, D_{overlaps} + N_{overlaps}, C - (D_{overlaps} + N_{overlaps}), D, lower.tail = TRUE)$. Following the hypergeometric test, $P$ values were adjusted for multiple comparisons using the Benjamini-Hochberg procedure based on the number of annotations tested per dataset. Summary plots of methylation were generated using the plotAnnopeak function of ChIPseeker and Heatmaps of DMRs were generated using ComplexHeatmap[103].

HP1cTKO RRBS data was quality trimmed using trim galore using the --rrbs flag and prepared for analysis using Bismark[104] with bowtie1 and arguments -n 1 -l 45. Bismark coverage files from WT and HP1cTKO were analysed using the R package methylKit[102], where all differentially methylated bases were extracted that change more than 25% and pass a corrected significance threshold of q = 0.01. Differentially methylated bases were then annotated using genomation[105] and prepared bed files (CpGs, LINEs, SINEs, LTRs, Exons) retrieved from the UCSC table browser. Jitter plots were created using ggplot2. Circos plot was prepared by using the circlize package[106]. HP1cTKO eAge was calculated by using the -18,000 CpGs included in the MouseEpigeneticClock tool (https://github.com/EpigenomeClock/MouseEpigeneticClock)[107]. In the WT sample 966 (5.36%) sites were imputed whereas in HP1cTKO 946 (5.25%) sites were imputed. The original eAge prediction for WT was -0.9 weeks and the HP1cTKO 0 weeks resulting in a positive difference of 0.9 weeks (6.3 days).

### ChIPseq

ChIP raw data was quality filtered using Trimmomatic and aligned to GRCm38.p5 using bowtie. Deeptools was used to generate bigwig files and generate heatmaps and profiles[96]. RPGC normalized bigwig files were created for each biological replicate using bamCompare with parameters --normalizeUsing RPGC--effectveGenomeSize 2652783500 --operation mean --extendReads 125. Two biological replicate bigwigs were then averaged using bigwigCompare with --operation mean. For IAPEz heatmaps, a comatrix was then computed from normalized averaged bigwigs over *IAPEz-int* regions using the computeMatrix scale-regions with --binsize 50 --regionBodyLength 1000 --beforeRegionStartLength 1000 --afterRegionStartLength 1000. Heatmap was then plotted using plotHeatmap with --colorMap Greens Blues Oranges--heatmapHeight 10--heatmapWidth 3--zMax 5. KAP1, H3K9me3 and H4K20me3 binding profiles at ICRs were generated by creating a center-point comatrix at ZFP57 peaks overlapping published murine ICRs with computeMatrix reference-point --binSize 150 --referencePoint center --beforeRegionStartLength 3000 --afterRegionStartLength 3000. For this annotation, previously published ZFP57 peaks[108] were subset by their overlap with published murine ICRs.

### Reporting summary

Further information on research design is available in the Nature Portfolio Reporting Summary linked to this article.

## Data availability

The sequencing data from this study has been deposited at GEO database under accession # GSE153331. ZFP57 Binding sites were obtained from GSE77444. Raw Data from figures and statistics from this work can be found in the Source Data file. Source data are provided with this paper.

## Code availability

Code used in analysis can be accessed at https://github.com/qoldt/HP1-Deficiency-Neurodegeneration.

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

## Acknowledgements

This work was supported by DFG research grant 410579311 (A.G.N. & V.T.); During revision of the manuscript, V.T. was supported by the Ministry of Science and Higher Education of the Russian Federation (project no. FSWR-2023-0029). Funding to S.Z. was provided by the German Research Foundation (DFG, SFB665, SFB1315), and the German Academic Exchange Service (DAAD); Funding to P.B-S. (P. Boehm-Sturm) was provided by the German Federal Ministry of Education and Research (BMBF) under the ERA-NET NEURON scheme (01EW1811) and the DFG (research grant BO 4484/2-1, 424778381-TRR 295 ReTune and EXC-2049-390688087 NeuroCure). Research by P.B.S. (P. B. Singh) was supported by a grant of the Ministry of Education and Science of the Republic of Kazakhstan, № AP19678932 under the Ministry of Education and Science of the Republic of Kazakhstan for "Regulation of Cellular Identity and Plasticity by mammalian HP1 proteins" program; funding was also by the Ministry of Health of the Republic of Kazakhstan under the program-targeted funding of the Ageing and Healthy Lifespan research program (IRN: 51760/ПЦФ-МЗ РК–19). Computation was performed on the HPC for Research/Clinic cluster of the Berlin Institute of Health. We would like to acknowledge Julie Brind'Amour, Carol Chen and Matthew Lorincz for early discussions. We thank Marion Rivalan and Melissa Long from the NeuroCure Animal Outcome Core Facility (AOCF) for facilitating behavioral experiments. We thank Robert Schwarz for statistical advice on calculating Odds Ratios. We thank T. Nowakowski for critically reading the manuscript draft. We thank K. Horie for the pFL IAP plasmid template. We also thank Prof. Dr. Seija Lehnardt and Christina Krüger for advice with astrocyte culture. We would also like to thank Ingo Bormuth, Roman Wunderlich, Ulrike Günther, Denis Lajkó and the animal facility of the Charité.

## Author contributions

Conceptualization, A.G.N., P.B.S. and V.T.; Investigation, A.G.N., J.S., S.Z., P.B., Sh.Ma., R.D., D.R., P.B-S.; MRI, Su. Mu., P.B-S., Resources, A.G.N., J.P.B., M.N., O.O., H.K.; Cloning of HP1 expression constructs and In situ probes, A.G.N.; In utero electroporations, A.G.N; Cloning of IAP T7 Construct, T.S.; Bioinformatics, A.G.N.; Mixed glial cultures and biochemistry, D.R., Formal analysis, A.G.N.; Visualisation, A.G.N.; Funding acquisition, A.G.N., P.B.S., V.T.; Writing-original draft, A.G.N., Writing-review and editing, A.G.N., P.B., P.B.S., V.T., Writing-revision A.G.N., P.B.S.; All authors reviewed the final manuscript.

## Funding

## Competing interests

The authors declare no competing interests.
