## [Transparent Peer Review file · Nature Communications]

Glial Reactivity and Cognitive Decline Follow Chronic Heterochromatin Loss in Neurons

Corresponding Author: Professor Victor Tarabykin

Version 0:

Reviewer comments:

Reviewer #1

(Remarks to the Author)

In this study, Newman and colleagues investigate the role of HP1 deficiency in neurodegeneration using mouse model studies. The authors demonstrate that forebrain-specific deletion of HP1 and HP1 leads to the loss of heterochromatin, alterations in DNA methylation, and widespread transcriptional increase in endogenous retroviruses (ERVs), the repetitive elements that are heterochromatically silenced under normal physiological conditions. With substantial characterizations and analyses, the authors show that HP1 /HP1 double knockout (DKO) animals exhibit age-dependent abnormalities, including glial and complement activation, CA3 dendritic degeneration, reduced brain volume, behavioral deficits, and cognitive impairment.

Overall, the authors provide a significant body of work underscoring the importance of HP1s and heterochromatin maintenance in the nervous system. While I generally support publication of the manuscript, it still remains unclear whether these correlative neuropathological deteriorations are causal to ERVs activation in the DKO mice. Additional experiments should be performed to address the points below and clarify some of the concepts/statements raised in the manuscript prior to publication.

Specific points:

(1) There is no data indicating that increased ERVs are a major cause of neurodegeneration via the complement pathway in DKO mice. The experiments that elucidate the causality of ERVs activation and neurodegeneration is important and is necessary to support the study's claims. For instance, could knockdown of ERVs in neurons attenuate glial/complement activation and rescue neurodegenerative phenotypes in the DKO mice? Or could overexpression of ERVs in wild-type neurons lead to similar pathological consequences that were observed in the DKO mice?

(2) The experiments/analyses were conducted using either brain samples from DKO mice or lysates of ES cells (ESCs) with HP1, HP1, and HP1 triple KO (TKO). Could the authors explain the rationale of using TKO ESCs instead of DKO ESCs? Given that different cell types have unique and specific DNA methylation patterns, the authors should perform the same experiments to confirm the DNA methylation changes also occur in neuronal cells lacking HP1s.

(3) In Extended Data Figure S7b, the C3 signal seems very robust in the hippocampal CA3 region. Does the signal come from neurons? In addition, the quantification of C3+ astrocytes and CD68+ microglia should be included in Figure 3 and extended data Figure S7.

(4) Figure S4 is not novel. The interaction between HP1 and SUV420H2 has been reported previously. (Reference: Souza et al., 2009, BMC Cell Biology.)

(5) The statements listed below are confusing or incorrect and need to be clarified:

- Line 52: No evidence indicates an infiltration of microglia in DKO mice.
- Line 93: This sentence is not correct. TDP-43 and Tau are not risk factors for neurodegenerative diseases. Instead, mutations in TDP-43 or Tau are associated with neurodegeneration. Moreover, Ref 20 does not suggest elevated levels of ERVs in patients with FTL.

- Extended Data Figure S2c, it's unclear if the scale bar (count) on the right belongs to this panel.
- Line 166: The first are dentate gyrus related genes underrepresented in HP1 β KO and HP1 β/γ DKO (140). What are those genes that are related to the dentate gyrus?
- Line 211: repeats LINEs and LTRs show hypomethylation not hypermethylation in HP1cTKO ESCs.
- Line 236: We found that cytosine methylation at Neuronatin (Nnat) is decreased in an age dependent manner in neurons deficient for HP1 β , while a trend towards hydroxymethylation can be observed in HP1 β/γ DKO neurons (fig. 2C). Similarly, DNA methylation at IAP sequences showed an age dependent decrease in HP1 β/γ DKO neurons (fig. 2C). The authors should replace "neurons" with "hippocampal lysates" as these assays were conducted in hippocampal tissue, where multiple cell types are present.
- Line 418: Age-related loss of maintenance methylation appears to be restricted to loci already under threat of activation such as the silenced allele of a tissue-specific imprinted gene (i.e., Nnat) or partially silenced repeats such as IAPs. Without allele-specific analysis, the authors do not have evidence to state the activation of the silenced allele.

Reviewer #2

(Remarks to the Author)

In this manuscript A.G. Newman et al. examined the conditional deletion of HP1 genes in mouse forebrains and found that loss of HP1 proteins causes impairment of heterochromatin maintenance and de-repression of ERVs including retrotransposons intracisternal A-particles (IAPs). By analyzing single and double conditional knockouts of HP1 β and HP1 γ genes, the authors found that HP1 γ is necessary and sufficient for H4K20me3 deposition and HP1 β is required for DNA maintenance methylation. Progressive ERV de-repression in HP1 β/γ DKO mice is correlated with the induction of complements and immune pathway genes as well as the increased infiltration and activation of microglia in the cortex, which is associated with the decrease in dendritic complexity and the decline in cognitive behaviors. The results are of general interests to the readers of Nature Communications. Nevertheless, the manuscript should be further revised by addressing the following concerns:

Major concerns:

1. There is clear evidence that the loss of heterochromatin causes the global reactivation of the repeat elements. However, the authors should tone down their conclusion that de-repression of ERVs results in neurodegeneration. The authors would need to provide more data about the relationship of the activation of complement cascade and other immune genes with ERV reactivation upon HP1 deficiency. ERV reactivation could be an independent event in parallel to the elevation of complement cascade. The current data in this study only supports an association of ERV reactivation with neurodegenerative phenotype, but not the causal relationship yet.
2. The relationship between HP1 deficiency and DNA methylation changes is not well described in this study. The authors performed RRBS in control and mutants and argued that 5hmC is probably involved in DNA methylation changes as a consequence of HP1 deficiency. The author may need to provide additional genome-scale analysis of 5hmC together with RRBS data. This is particularly important to address whether IAP reactivation is actually secondary to DNA methylation changes after HP1-deficiency induced changes in histone modifications.
3. In the line 233, the authors claim "HP1 β deficiency results in decay of DNA maintenance methylation". Indeed, both hyper and hypo methylated sites could be identified due to the heterochromatin loss. But it is not clearly enough to show this is the result of decay of DNA maintenance methylation. The hyper and hypo methylated sites could result from the de novo methylation (hypermethylation) and demethylation (hypomethylation) as a consequence to the heterochromatin loss.

Minor points:

1. The capital and lower-case letter is not consistent in the figure and figure legends. Please check FigS4 as an example.
2. For the Extended Data Fig 1c, how many samples were used for Statistical analysis?
3. For the Extended Data Fig S2C, is it possible to figure a better way for visualization? In the current Fig S2C, the up-regulation of ERVs seems not dramatic.
4. Does the abnormal DNA methylation show any preference on the genomic regions? The author observed abnormal DNA methylation in the ICRs. Hypergeometric test is needed to confirm the abnormal DNA methylation tend to occur in ICRs instead of a result from random distribution.

Reviewer #3

(Remarks to the Author)

In this study, Newman et al., investigated the role of HP1 β and HP1 γ in the reactivation of endogenous retroviruses (ERV) in mouse brain and propose that "de-repression of ERVs results in neurodegeneration via the Complement cascade in an

age dependent manner". Data used in support of the claims involve IHC, RNA-seq, DNA methylation, in-situ hybridization and behavioral experiments in mouse. Molecular analyses were also performed in ESC cells (such as ChIP-seq).

The study is interesting, however there are no data to support the causal relationship between ERV reactivation, the Complement cascade and neurodegeneration. Rescue experiments should be performed, for instance, by knocking down ERV in HP1 β KO mice, thus, to show repression of the Complement cascade. Similarly, a direct relationship between the Complement cascade and neurodegeneration should be demonstrated. Given that the HP1 β and HP1 γ KO was done in both neurons and glia, and that neurons and glia play different functions in the brain, the authors should justify why taking a non-cell-type specific approach.

Major comments on the data analysis:

In-situ hybridization and IHC: There is no quantification on the images, nor reporting on number of mice/tissue sections analyzed.

"Data S1": Couldn't be found among the submission material.

GSEA analysis: There is no info on the parameters used for this analysis. This analysis should be accompanied by an unbiased Gene Ontology (GO) analysis to show that similar functional categories are involved.

ChIP-seq: ChIP-seq has been performed without Input normalization. Refer to the ENCODE guidelines on ChIP-seq experiments and Input normalization.

H4K20me3 ChIP-seq: This data should be generated in the mouse brain for direct correlation with gene expression changes (e.g. loss of H4K20me3 correlating with ERV is derepressed; and similarly for the Complement cascade genes).

Figure legends: need more details and clearer description of the figure panels.

Version 1:

Reviewer comments:

Reviewer #1

(Remarks to the Author)

The authors have performed additional experiments and addressed all the comments. However, one concern remains with the new experiments in Figure 6. While the use of in vitro transcribed IAP RNA to test the causal role of ERV derepression is a step in the right direction, this experimental design does not fully distinguish between general RNA immunogenicity and ERV-specific effects. The suppression of immune activation by pseudo-uridine is a well-known phenomenon and is not an appropriate control for isolating ERV-specific effects. To more conclusively demonstrate that ERV RNA has enhanced immunomodulatory properties compared to other RNAs, the authors should use a scrambled RNA control with the same length and nucleotide composition as the IAP RNA.

Reviewer #2

(Remarks to the Author)

The authors have performed additional experiments to address most of the concerns of the reviewers. With regards to over-expression of IAP RNA in cells, I am curious about what is the effect of IAP RNA over-expression in hippocampal neurons, perhaps via RNA electroporation. Perhaps it is an interesting experiment for the next paper. In summary, I feel that the revision has made the paper acceptable to be published in Nature Communications.

Reviewer #3

(Remarks to the Author)

The authors have conducted additional experiments and analyses to address the reviewers' comments, including the exogenous administration of IAP ssRNA in astrocyte-microglia co-culture, and new experiments on 5mC and 5hmC. While the IAP ssRNA administration does activate microglia and astrocytes, it is still unclear whether the reactivation of ERVs is specific to neurons or if it also extends to astrocytes and microglia due to HP1 β and HP1 γ KO, which could indicate a cell-autonomous mechanism – could the authors provide evidence that there is no ERV activation in microglia and astrocyte upon HP1 KO?

The authors state in the reviewer's response that activation of microglia and astrocytes upon IAP ssRNA administration may not operate on the Complement C3 - this is not fully addressed in the revised manuscript.

The relevance of the unfolded protein response (UPR) and complement cascade in HP1 KO mice remains ambiguous in the RNA-seq data, as these are not prominent in the top GO terms, which instead highlight cell adhesion and synaptic processes. Although identified through GSEA analysis, the specific gene sets used and the significance of enrichment are not presented; instead, network nodes are generated with Cytoscape. Furthermore, the functional relevance of upregulation of Protocadherin genes in the HP1 γ KO has not been discussed (cell adhesion is one of the top GO terms in the RNA-seq

results).

Overall, the presentation of the RNA-seq data is somewhat confusing, and the selection of genes for follow-up doesn't always seem statistically driven.

Specific comments are as follows:

- Data S1/Table S1 lacks a list of differentially expressed genes (DEGs) and ERVs along with their expression values and q-values.
- It is unclear how many endogenous retroviruses (ERVs) are significantly differentially expressed.
- Sequencing alignment statistics for RNA-seq and ChIP-seq libraries are missing.
- The ChIP-seq protocol requires a more detailed description, including any statistical analyses used to detect differentially enriched genomic regions.
- Some figures and figure legends need revision, for instance, Figure 1c lacks a scale bar, and several figure legends describe the results/interpretation of the data without providing accurate description of the data (e.g. Fig. s1f).

Version 2:

Reviewer comments:

Reviewer #1

(Remarks to the Author)

The authors have addressed all of the concerns raised by this reviewer. I support the publication of the study in Nature Communications.

Reviewer #3

(Remarks to the Author)

The authors have addressed several of the previous comments, including by providing clarifications in the text, expanding the discussion, incorporating GSEA plots, and missing supplementary information.

On the issue of cell type-specific ERV activation, the authors state that while HP1 proteins are deleted from both neurons and astrocytes, ERV expression is not observed in astrocytes (e.g., Fig. 5d,f). However, these panel don't clearly show astrocyte-specific co-staining with IAPs or other ERVs, and the resolution is insufficient to support this conclusion. This point should either be addressed with higher-resolution images and quantification or clearly acknowledged as a limitation, as ERV expression in astrocytes would significantly affect the overall interpretation of the findings.

Regarding IAP ssRNA-induced activation of microglia and astrocytes, the authors state that no C3 changes were detected by Western blot at 24 or 48 hours ("data not shown"). These data should be included in supplementary information, and if the model doesn't recapitulate the in vivo environment, this should be noted as a limitation.

The ChIP-seq experimental methods section remains limited. Inspection of the GEO dataset suggests no Input control was used, which limits confidence in differential enrichment results. This omission should be acknowledged as a limitation in the manuscript.

Finally, Fig. 5n (iii) appears out of focus and should be reviewed or replaced.

Overall, while some points have been addressed, some claims, particularly regarding cell type specificity and C3 activation, require either additional data or clearer acknowledgment of limitations.

new title:

Glial Reactivity and Cognitive Decline Follow Chronic Heterochromatin Loss in Neurons

REVIEWER COMMENTS

Reviewer #1 (Remarks to the Author):

In this study, Newman and colleagues investigate the role of HP1 deficiency in neurodegeneration using mouse model studies. The authors demonstrate that forebrain-specific deletion of HP1 β and HP1 γ leads to the loss of heterochromatin, alterations in DNA methylation, and widespread transcriptional increase in endogenous retroviruses (ERVs), the repetitive elements that are heterochromatically silenced under normal physiological conditions. With substantial characterizations and analyses, the authors show that HP1 β /HP1 γ double knockout (DKO) animals exhibit age-dependent abnormalities, including glial and complement activation, CA3 dendritic degeneration, reduced brain volume, behavioral deficits, and cognitive impairment.

Overall, the authors provide a significant body of work underscoring the importance of HP1s and heterochromatin maintenance in the nervous system. While I generally support publication of the manuscript, it still remains unclear whether these correlative neuropathological deteriorations are causal to ERVs activation in the DKO mice. Additional experiments should be performed to address the points below and clarify some of the concepts/statements raised in the manuscript prior to publication.

Specific points:

(1) There is no data indicating that increased ERVs are a major cause of neurodegeneration via the complement pathway in DKO mice. The experiments that elucidate the causality of ERVs activation and neurodegeneration is important and is necessary to support the study's claims. For instance, could knockdown of ERVs in neurons attenuate glial/complement activation and rescue neurodegenerative phenotypes in the DKO mice? Or could overexpression of ERVs in wild-type neurons lead to similar pathological consequences that were observed in the DKO mice?

We agree, until now the evidence we provided did not provide a direct pathway from ERV expression and complement activation. In considering experimental options, it was difficult to envision a method that would ensure the knockdown of ERVs at the transcriptional level. However we did think an over-expression based experiment could in theory show ERV directed activation of complement. Given our observation that ERV RNA is de-repressed from neurons but not astrocytes, and that the main difference we can observe is the presence of C3+ reactive astrocytes in aged HP1DKO hippocampi, we designed a new set of experiments to test the effect of IAP ERV RNA on mixed glial cultures (new figure 6). For these experiments we cloned the sequence of the full length IAP and transcribed RNA in vitro. For an experimental control we used the same IAP sequence template but instead used pseudo-uracil (ψ -IAP) during in vitro transcription.

Here we could observe that a single exogenous application of IAP ssRNA, but not ψ -IAP resulted in acute activation of inflammatory pathways in a mixed astrocyte and microglial culture (new figure 6), where we could observe acute inflammatory response similar to IGN- γ stimulation. We were not able to observe a direct effect on Complement 3 from this acute paradigm, however the in vivo environment is considerably more complex and exists as a chronic inflammatory environment over the mouse's lifetime (15 months). Accordingly, we have moderated the language surrounding ERV-directed activation of Complement 3, and have also accordingly changed the title of the Manuscript to reflect this. This has also been reflected in a newly-written discussion, which discusses our results in the context of recently published studies in this area.

(2) The experiments/analyses were conducted using either brain samples from DKO mice or lysates of ES cells (ESCs) with HP1 α , HP1 β , and HP1 γ triple KO (TKO). Could the authors explain the rationale of using TKO ESCs instead of DKO ESCs? Given that different cell types have unique and specific DNA methylation patterns, the authors should perform the same experiments to

confirm the DNA methylation changes also occur in neuronal cells lacking HP1s.

We thank the reviewer for this question, it is true, different cell types including neurons have unique DNA methylation patterns as well as differing hydroxymethylation. Initially we sought to examine the effect of loss of HP1 proteins on DNA methylation at repetitive elements using HP1cTKO cells to avoid any compensation from HP1a, and could indeed observe strong effects. Given the importance of cell type and the unique methylation and 5-hydroxymethylation that occurs in neurons, we took this opportunity to perform a new experiment to profile 5mC and 5hmC in hippocampi from all genotypes and ages. The results of this experiment confirmed that DNA methylation is lost at repeats in HP1 mutant neurons and the results of this experiment can be found in the new figure 4.

(3) In Extended Data Figure S7b, the C3 signal seems very robust in the hippocampal CA3 region. Does the signal come from neurons? In addition, the quantification of C3+ astrocytes and CD68+ microglia should be included in Figure 3 and extended data Figure S7.

This is correct there also appears to be strong C3 signal in the CA3 region corresponding to the soma of the pyramidal Neurons there. This appears to be a general age-dependent affect in complement expression in this region, which is also consistent with newly published spatial transcriptomics data (Hahn et al., 2023 Cell). In aged WT and single mutants we did not observe any C3+ astrocytes in the stratum radiatum. We have additionally added a quantification of CD68+ microglia in (new figure 5m), and added explicit mention of the C3+ signal in neurons in the main text:

“Elevated C3 transcripts could be observed in the soma of wildtype CA1 and CA3 neurons, but in aged HP1 β/γ DKO, C3 could also be detected in small plaque-like foci in the stratum radiatum (white arrows, fig. 5d,g) that were not observed in any other condition and surrounded by Iba1+ microglia with a distinct morphology (fig. 5e,f).”

(4) Figure S4 is not novel. The interaction between HP1 γ and SUV420H2 has been reported previously. (Reference: Souza et al., 2009, BMC Cell Biology.)

We have included mention of the 2009 paper to the results:

“Given the known requirement of the HP1 chromoshadow domain (CSD) for association with the H4K20me3 HMTase Suv420h2 (Souza et al., 2009), we carried out co-immunoprecipitation and co-localization analysis to identify residues in the CSD essential for the interaction of HP1 γ with Suv420h2 (fig. 2c-e, S4).”

(5) The statements listed below are confusing or incorrect and need to be clarified:

- Line 52: No evidence indicates an infiltration of microglia in DKO mice.

We have clarified this point to refer to the fact that microglia are increased in the hippocampus of aged HP1DKO hippocampi as per figure 3j (new figure 5j,k).

- Line 93: This sentence is not correct. TDP-43 and Tau are not risk factors for neurodegenerative diseases. Instead, mutations in TDP-43 or Tau are associated with neurodegeneration. Moreover, Ref 20 does not suggest elevated levels of ERVs in patients with FTLN.

We amended the text to refer to the correct citation which now reads as follows:

“Elevated levels of ERVs have also been seen in models examining factors associated with neurodegenerative diseases such as Tau²¹ and TDP-43²², while α -synuclein has been shown to affect chromatin and the maintenance of ERVs directly^{23,24}”

- Extended Data Figure S2c, it's unclear if the scale bar (count) on the right belongs to this panel.

This scale bar does indeed belong with this plot, it was a hexplot designed to mediate any overplotting at areas with many observations. Given a similar point raised by another reviewer, we have changed the plot to a linear scale and converted to normal points.

- Line 166: The first are dentate gyrus related genes underrepresented in HP1 β KO and HP1 β / γ DKO (140). What are those genes that are related to the dentate gyrus?

We have included mention of a few examples in the text, and referred to the expression table where all genes downregulated in HP1DKO/BKO can be found:

"...dentate gyrus related genes underrepresented in HP1 β KO and HP1 β / γ DKO (140) such as *Prox1*, *Dsp*, *Plk5* and *Cdh9* (Data S1)."

- Line 211: repeats LINEs and LTRs show hypomethylation not hypermethylation in HP1cTKO ESCs.

We regret this error, & thank the reviewer for catching this. This line has been corrected to be:

"HP1cTKO ESCs displayed hypomethylation at LINEs and LTRs (particularly ERVK) (fig. 3d,g) despite a global shift towards hypermethylation (fig.3e) which is also represented in promoters and gene bodies (fig 3f). "

- Line 236: We found that cytosine methylation at Neuronatin (*Nnat*) is decreased in an age dependent manner in neurons deficient for HP1 β , while a trend towards hydroxymethylation can be observed in HP1 β / γ DKO neurons (fig. 2C). Similarly, DNA methylation at IAP sequences showed an age dependent decrease in HP1 β / γ DKO neurons (fig. 2C). The authors should replace "neurons" with "hippocampal lysates" as these assays were conducted in hippocampal tissue, where multiple cell types are present.

We have corrected both of these statements to reflect their source:

"We found that cytosine methylation at Neuronatin (*Nnat*) is decreased in an age-dependent manner in HP1 β fl/fl *Emx1Cre* hippocampal lysates. Similarly, HP1 β fl/flHP1 γ fl/fl *Emx1Cre* hippocampal lysates showed age-dependent decreases in 5mC methylation at IAP sequences alongside a trend towards hydroxymethylation for both loci (fig. S6a)."

- Line 418: Age-related loss of maintenance methylation appears to be restricted to loci already under threat of activation such as the silenced allele of a tissue-specific imprinted gene (i.e., *Nnat*) or partially silenced repeats such as IAPs. Without allele-specific analysis, the authors do not have evidence to state the activation of the silenced allele.

This is true. We have modified the statement to read as follows:

"Age-related 5mC hypomethylation appears to be restricted to loci already under threat of activation such as the tissue-specific imprinted genes (i.e., *Nnat*) or partially silenced ERVK repeats such as IAPs."

Reviewer #2 (Remarks to the Author):

In this manuscript A.G. Newman et al. examined the conditional deletion of HP1 genes in mouse forebrains and found that loss of HP1 proteins causes impairment of heterochromatin maintenance and de-repression of ERVs including retrotransposons intracisternal A-particles (IAPs). By analyzing single and double conditional knockouts of HP1 β and HP1 γ genes, the authors found that HP1 γ is necessary and sufficient for H4K20me3 deposition and HP1 β is required for DNA maintenance methylation. Progressive ERV de-repression in HP1 β / γ DKO mice is correlated with the induction of complements and immune pathway genes as well as the increased infiltration and activation of microglia in the cortex, which is associated with the decrease in dendritic complexity

and the decline in cognitive behaviors. The results are of general interests to the readers of Nature Communications. Nevertheless, the manuscript should be further revised by addressing the following concerns:

Major concerns:

1. There is clear evidence that the loss of heterochromatin causes the global reactivation of the repeat elements. However, the authors should tone down their conclusion that de-repression of ERVs results in neurodegeneration. The authors would need to provide more data about the relationship of the activation of complement cascade and other immune genes with ERV reactivation upon HP1 deficiency. ERV reactivation could be an independent event in parallel to the elevation of complement cascade. The current data in this study only supports an association of ERV reactivation with neurodegenerative phenotype, but not the causal relationship yet.

We agree, until now the evidence we provided did not provide a direct pathway from ERV expression and complement activation. In considering experimental options, it was difficult to envision a method that would ensure the knockdown of ERVs at the transcriptional level. However we did think an over-expression based experiment could in theory show ERV directed activation of complement. Given our observation that ERV RNA is de-repressed from neurons but not astrocytes, and that the main difference we can observe is the presence of C3+ reactive astrocytes in aged HP1DKO hippocampi, we designed a new set of experiments to test the effect of IAP ERV RNA on mixed glial cultures (new figure 6). For these experiments we cloned the sequence of the full length IAP and transcribed RNA in vitro. For an experimental control we used the same IAP sequence template but instead used pseudo-uracil (ψ -IAP) during in vitro transcription.

Here we could observe that a single exogenous application of IAP ssRNA, but not ψ -IAP resulted in acute activation of inflammatory pathways in a mixed astrocyte and microglial culture (new figure 6), where we could observe acute inflammatory response similar to IGN- γ stimulation. We were not able to observe a direct effect on Complement 3 from this acute paradigm, however the in vivo environment is considerably more complex and exists as a chronic inflammatory environment over the mouse's lifetime (15 months). Accordingly, we have moderated the language surrounding ERV-directed activation of Complement 3, and have also accordingly changed the title of the Manuscript to reflect this. This has also been reflected in a newly-written discussion, which discusses our results in the context of recently published studies in this area.

2. The relationship between HP1 deficiency and DNA methylation changes is not well described in this study. The authors performed RRBS in control and mutants and argued that 5hmC is probably involved in DNA methylation changes as a consequence of HP1 deficiency. The author may need to provide additional genome-scale analysis of 5hmC together with RRBS data. This is particularly important to address whether IAP reactivation is actually secondary to DNA methylation changes after HP1-deficiency induced changes in histone modifications.

We agree entirely with the reviewer and have performed a new set of experiments with the intent to separate 5mC and 5hmC in HP1DKO hippocampal lysates. We utilized the Tecan Ovation system which is able to profile both 5mC and 5hmC, and have put the results of this experiment in the new figure 4. We did indeed find that in HP1DKOs, DNA methylation at ERVK elements was lost, and we found age-related increases in 5hmC at several non-coding elements.

3. In the line 233, the authors claim "HP1 β deficiency results in decay of DNA maintenance methylation". Indeed, both hyper and hypo methylated sites could be identified due to the heterochromatin loss. But it is not clearly enough to show this is the result of decay of DNA maintenance methylation. The hyper and hypo methylated sites could result from the de novo methylation (hypermethylation) and demethylation (hypomethylation) as a consequence to the

heterochromatin loss.

This section has been extensively re-written given the new 5mC and 5hmC results from the new experiment. It now reads as follows:

“ Prominent increases in the expression of tissue specific imprinted genes (Data S1) in HP1 β KO and HP1 β/γ DKO mutants indicated that DNA methylation may be affected by HP1 β deficiency.

We engineered an ES cell line that contains an ERT2-Cre transgene where all three HP1 genes are floxed. This system allowed for the deletion of all three HP1 genes following the addition of tamoxifen, (thus termed HP1cTKO) that we then be profiled for DNA methylation using reduced-read bisulfite sequencing (RRBS) and changes to KAP1, H3K9me3 and H4K20me3 (Figure 3a). We found that triple deficiency of HP1 proteins results in reduced KAP1 at imprinting control regions (ICRs) marked by ZFP57 (fig. S5a), while DNA methylation at these ICRs including Nnat shows mixed changes (fig. S5b,c). KAP1 was unchanged over IAP elements in HP1cTKO ES cells, with a reduction in H3K9me3 and an expected absence of H4K20me3 (fig. 3b). While ES cells are not expected to utilize protocadherins the same way as neurons, regulatory H3K9me3 in the cPcdh is unaffected in HP1cTKO ESCs despite H3K9me3 and H4K20me3 being lost at adjacent IAP elements (fig S5d, arrows). This additionally suggests that regulatory H3K9me3 is unaffected in HP1 γ fl/fl Emx1Cre mutants, and that it is disruption of the HP1 γ -H4K20me3 pathway that causes elevated Pcdh expression in HP1 γ fl/fl Emx1Cre and subsequently HP1 β fl/fl HP1 γ fl/fl Emx1Cre brains.

DNA methylation can be used to accurately estimate the biological or ‘epigenetic’ age (eAge) of tissues, which increases at a steady rate in adult tissues ⁷. While embryonic stem cells normally maintain an assigned eAge of zero ⁷, we were surprised to find that the eAge of HP1cTKO ES cells increased faster than their time in culture (fig. 3c).

HP1cTKO ESCs displayed hypomethylation at LINEs and LTRs (particularly ERVK) (fig. 3d,g) despite a global shift towards hypermethylation (fig.3e) which is also represented in promoters and gene bodies (fig 3f). While a large number of significantly hypermethylated cytosines overlap with CpGs, exons, SINES (fig 3d) and ICRs (fig S5b), as annotation sets they do not show statistical significance based on the hypergeometric test (Fig. 3g). This lack of significance may be attributed to the global background shift towards hypermethylation.

Given that reduced representation bisulfite sequencing (RRBS) cannot discriminate between 5-methyl cytosine (5mC) and 5-hydroxy-methyl cytosine (5hmC), we surmised that some of the ‘hypermethylation’ observed in the HP1cTKO RRBS samples may be attributable to active demethylation processes, specifically the oxidation of 5mC to 5hmC mediated by TET enzymes.⁴³ We tested two specific loci using primers designed to amplify over single HpaII/MspI (CCGG) sites from hippocampal lysates. We found that cytosine methylation at Neuronatin (Nnat) is decreased in an age-dependent manner in HP1 β fl/fl Emx1Cre hippocampal lysates. Similarly, HP1 β fl/flHP1 γ fl/flEmx1Cre hippocampal lysates showed age-dependent decreases in 5mC methylation at IAP sequences alongside a trend towards hydroxymethylation for both loci (fig. S6a).

To elucidate the full effects of HP1 deficiencies on 5mC and 5hmC, we conducted RRBS both with and without prior oxidation on hippocampal lysates. This oxidation step converts 5hmC into 5-formylcytosine (5fC), which does not undergo bisulfite conversion, thereby allowing us to distinguish between 5mC and 5hmC (fig 4a). After an average 67% unique (+20% ambiguous) read mapping alignment efficiency we found that bisulfite conversion using this method recovered on average ~35% of Methylated CpGs (fig 4b). This experiment yielded results largely consistent with what was observed in HP1cTKO ESCs. We found that deficiency of HP1 β results in aberrant 5mC (de novo) hypermethylation of promoters and CpG islands, a large percentage coming from a gene desert on chromosome 12 (fig 4c-e, S6c). Surprisingly, we also found that clustered protocadherin promoters became dramatically 5mC hypomethylated in HP1 β/γ DKO hippocampi (fig 4e, S6d)., along with imprinting control regions (ZFP57 ICRs) (fig 4e). 5mC DNA methylation at LTRs, primarily ERVK elements, was markedly reduced in HP1 β/γ DKO hippocampi (Figure 4e, S6e). This effect is also likely an underestimate given the lower mapability of repeats in RRBS sequencing and a lower bisulfite conversion efficiency in this experiment. We could also observe corresponding statistically significant increases in 5hmC over ERVK and IAP/LTR1a annotated regions in HP1 β/γ DKOs. Notably, single deficiency of either HP1 β or HP1 γ already initiates drift in DNA methylation, evidenced by significant increases in 5hmC over introns, LINEs, SINES, and LTRs, an effect that is recapitulated in normal aging in wildtype hippocampi (fig. 4e).

Minor points:

1. The capital and lower-case letter is not consistent in the figure and figure legends. Please check FigS4 as an example.

We appreciate the reviewer's keen eye. We have re-checked all figures for capitalization consistency.

2. For the Extended Data Fig 1c, how many samples were used for Statistical analysis?

We have ensured the number of samples used for statistical analysis for Supplemental figure 1c has been stated in the caption along with statistical method and post-hoc test. It now reads as follows:

"Dentate gyri of HP1 β KO and HP1 β / γ KO animals show malformation of the infrapyramidal blade due to mitotic exhaustion between postnatal day 0 and postnatal day 8. P0 WT n = 10, P0 HP1 β KO n = 4, P0 HP1 γ KO n = 7, P0 HP1DKO n = 11, P8 WT n = 4, P8 HP1 β KO n = 3, P8 HP1 γ KO n = 4, P8 HP1DKO n = 5. One way ANOVA with Dunnett's Multiple Comparison test where ** denotes $p < 0.05$, and * denotes $p < 0.01$."

3. For the Extended Data Fig S2C, is it possible to figure a better way for visualization? In the current Fig S2C, the up-regulation of ERVs seems not dramatic.

Given the comments of another reviewer regarding the clarity of this plot, we have replotted the values on a linear scale, with a tradeoff of hiding smaller values at low RPKM, chimeric transcripts in HP1DKO are now more visible.

4. Does the abnormal DNA methylation show any preference on the genomic regions? The author observed abnormal DNA methylation in the ICRs. Hypergeometric test is needed to confirm the abnormal DNA methylation tend to occur in ICRs instead of a result from random distribution.

We have included a new plot that includes the odds ratio and adjusted p values from the hypergeometric test over annotations measured in HP1cTKO RRBS (new figure 3) and from the new HP1 hippocampal 5mC and 5hmC data (new figure 4).

Reviewer #3 (Remarks to the Author):

In this study, Newman et al., investigated the role of HP1 β and HP1 γ in the reactivation of endogenous retroviruses (ERV) in mouse brain and propose that "de-repression of ERVs results in neurodegeneration via the Complement cascade in an age dependent manner". Data used in support of the claims involve IHC, RNA-seq, DNA methylation, in-situ hybridization and behavioral experiments in mouse. Molecular analyses were also performed in ESC cells (such as ChIP-seq).

The study is interesting, however there are no data to support the causal relationship between ERV reactivation, the Complement cascade and neurodegeneration. Rescue experiments should be performed, for instance, by knocking down ERV in HP1 β KO mice, thus, to show repression of the Complement cascade. Similarly, a direct relationship between the Complement cascade and neurodegeneration should be demonstrated. Given that the HP1 β and HP1 γ KO was done in both neurons and glia, and that neurons and glia play different functions in the brain, the authors should justify why taking a non-cell-type specific approach.

We agree, until now the evidence we provided did not provide a direct pathway from ERV expression and complement activation. In considering experimental options, it was difficult to envision a method that would ensure the knockdown of ERVs at the transcriptional level. However we did think an over-expression based experiment could in theory show ERV directed activation of complement. Given our observation that ERV RNA is de-repressed from neurons but not astrocytes, and that the main difference we can observe is the presence of C3+ reactive astrocytes in aged HP1DKO hippocampi, we designed a new set of experiments to test the effect of IAP ERV RNA on mixed glial cultures (new figure 6). For these experiments we cloned the sequence of the

full length IAP and transcribed RNA in vitro. For an experimental control we used the same IAP sequence template but instead used pseudo-uracil (ψ -IAP) during in vitro transcription.

Here we could observe that a single exogenous application of IAP ssRNA, but not ψ -IAP resulted in acute activation of inflammatory pathways in a mixed astrocyte and microglial culture (new figure 6), where we could observe acute inflammatory response similar to $\text{IGN-}\gamma$ stimulation. We were not able to observe a direct effect on Complement 3 from this acute paradigm, however the in vivo environment is considerably more complex and exists as a chronic inflammatory environment over the mouse's lifetime (15 months). Accordingly, we have moderated the language surrounding ERV-directed activation of Complement 3, and have also accordingly changed the title of the Manuscript to reflect this. This has also been reflected in a newly-written discussion, which discusses our results in the context of recently published studies in this area.

Major comments on the data analysis:

In-situ hybridization and IHC: There is no quantification on the images, nor reporting on number of mice/tissue sections analyzed.

We have quantified the amount of CD68+ area per microglia and included it in figure 5. And included n numbers for all quantifications in the figure captions. For ISH images we have additionally included mention of number of brains tested.

"Data S1": Couldn't be found among the submission material.

We apologize for the oversight, Data S1 has been sure to be included in the re-submission.

GSEA analysis: There is no info on the parameters used for this analysis. This analysis should be accompanied by an unbiased Gene Ontology (GO) analysis to show that similar functional categories are involved.

We have previously reported the GSEA parameters in the methods:

"Gene set enrichment and leading edge analysis was performed using GSEA GUI to query raw count data from aged HP1DKO and wildtype against c2.cp.reactome.v6.2.symbols.gmt using Signal2noise in gene_set mode with default parameters. "

We have additionally performed an unbiased Gene ontology analysis using clusterProfiler and included it in Figure S2 (e).

ChIP-seq: ChIP-seq has been performed without Input normalization. Refer to the ENCODE guidelines on ChIP-seq experiments and Input normalization.

These samples sequenced without sequencing the input as instead we opted for biological replicates and 1X normalization for each ChIP. Specifically, each ChIP was performed twice, using a biological replicate (separate cell line). ChIPseq reads were then 1X normalized (RPGC) and biological replicates averaged. We note that our H3K9me3, KAP1 & H4K20me3 ChIPseq signal in controls is comparable to published ChIPseq of these marks in mouse embryonic stem cells such as experiments GSE87041 and GSE84382.

H4K20me3 ChIP-seq: This data should be generated in the mouse brain for direct correlation with gene expression changes (e.g. loss of H4K20me3 correlating with ERV is derepressed; and similarly for the Complement cascade genes).

While we agree with the reviewer this would be of interest, within the scope of this study we were unable to produce a genetic system that would enable an experiment of H3K9me3/H4K20me3 ChIPseq in a pure population of HP1 knockout neurons. If we were able to do this experiment cleanly, given H4K20me3 appears to be entirely lost in HP1 γ KO neurons (Figure 2) we would expect a similar result to that seen in the HP1cTKO cells with loss of H4K20me3 at ERVs and globally. The possibility that H4K20me3 deposition is important for repression of complement

cascade genes is an interesting proposition, however one major point against this possibility is the lack of complement gene upregulation in HP1 γ KO hippocampi.

Figure legends: need more details and clearer description of the figure panels.

We have expanded all of the figure panel legends to include a more explicit description of their constituent panels.

** See Nature Portfolio's author and referees' website at www.nature.com/authors for information about policies, services and author benefits.

Glial Reactivity and Cognitive Decline Follow Chronic Heterochromatin Loss in Neurons

Authors: A.G. Newman^{1*}, J. Sharif², P. Bessa¹, S. Zaqout³, J. P. Brown^{1§}, D. Richter¹, R. Dannenberg¹, M., Nakayama⁴, S. Mueller^{5,6}, T. Schaub¹, S. Manickaraj¹, P. Böhm-Sturm^{5,6}, O. Ohara^{4,7}, H. Koseki², P.B. Singh^{1,8,#,*}, V. Tarabykin^{1,#,*}

RESPONSE TO REVIEWER COMMENTS

Reviewer #1 (Remarks to the Author)

The authors have performed additional experiments and addressed all the comments. However, one concern remains with the new experiments in Figure 6. While the use of in vitro transcribed IAP RNA to test the causal role of ERV derepression is a step in the right direction, this experimental design does not fully distinguish between general RNA immunogenicity and ERV-specific effects. The suppression of immune activation by pseudo-uridine is a well-known phenomenon and is not an appropriate control for isolating ERV-specific effects. To more conclusively demonstrate that ERV RNA has enhanced immunomodulatory properties compared to other RNAs, the authors should use a scrambled RNA control with the same length and nucleotide composition as the IAP RNA.

We agree that this was an oversight and have repeated this experiment with a newly synthesized 1000bp random (scr) RNA. We could also observe this RNA to be broadly immunogenic, however some of this could be because the Cytokine kit used to test scrRNA was newer, as this kit also showed higher sensitivity to the control. Despite the broad reactivity of scrRNA we still see high specificity of CCL5, CXCL10 and IL-23 in response to an acute stimulation with IAP ssRNA. We look forward to a future study where we can focus primarily on mapping this inflammatory reaction.

Reviewer #3 (Remarks to the Author)

The authors have conducted additional experiments and analyses to address the reviewers' comments, including the exogenous administration of IAP ssRNA in

astrocyte-microglia co-culture, and new experiments on 5mC and 5hmC. While the IAP ssRNA administration does activate microglia and astrocytes, it is still unclear whether the reactivation of ERVs is specific to neurons or if it also extends to astrocytes and microglia due to HP1 β and HP1 γ KO, which could indicate a cell-autonomous mechanism – could the authors provide evidence that there is no ERV activation in microglia and astrocyte upon HP1 KO?

We thank the reviewer for this careful consideration of the details. In this experimental system EMX1cre results in the deletion of HP1 proteins from neurons and astrocytes. However, from what we can observe, ERVs in HP1DKO brains only appear to be de-repressed in neurons, and seem to be more greatly de-repressed in pyramidal hippocampal neurons and the deep layer neurons of the cortex (Fig 1d, S1b, Fig 5). ERV de-repression is absent in the dentate gyrus and astrocytes. While we are still unsure as to why, other studies have observed that ERV transcription may be linked to Sox family transcription factors. We have included the following statement in the manuscript to make these points more declarative:

“We note that while astrocytes in an *Emx1^{Cre}* mutant will be deficient for HP1 proteins, we do not observe ERV expression in astrocytes (fig 5d,f). Rather, we observe ERV de-repression preferentially in the deep layer neurons of the cortex and hippocampal pyramids but not the dentate gyrus (fig 1d, S2b, 5). This suggests both that ERV re-activation requires cell-type specific transcription factors (such as previously suggested Sox-family transcription factors ¹), and that inflammatory effects on astrocytes and microglia are non cell-autonomous.”

The authors state in the reviewer’s response that activation of microglia and astrocytes upon IAP ssRNA administration may not operate on the Complement C3 - this is not fully addressed in the revised manuscript.

In addition to revising the description of the ssRNA experiment, we have included the following statement in the results section of the manuscript:

“Neither our 24hr or 48hr paradigms could detect changes in cytoplasmic C3 by Western blot (data not shown), however, our acute experimental protocol is unlikely to fully recapitulate the chronic inflammatory environment in the HP1DKO brain”

In the discussion we also hypothesize two pathways whereby complement activation might occur in astrocytes" First, via chronic immune upregulation of IFNGR2, which stabilizes C3 and C4 RNA and, second, by the known ability of ERV transcripts to induce the unfolded protein response, which itself can also lead to complement activation in astrocytes:

“When we tested the effect of non cell-autonomous ERV RNA on mixed glial cultures, we could observe some ERV RNA specific effects. The acute ERV ssRNA stimulation was followed by IRF7 phosphorylation and nuclear ingress (fig 6e-g). IRF7 activation occurs downstream of Pattern Recognition Receptors (PRRs) that recognize

ssRNA, either endosomal toll receptors TLR7/9, or cytoplasmic receptors such as RIG-1/MDA5⁶⁸. Once in the nucleus, phosphorylated IRF7 is known to directly activate type I interferons (IFN α/β) and Interferon Stimulatory Genes (ISGs)⁶⁹, including the cytokines CXCL10 and CCL5⁴⁸, which we robustly observe 24hrs and 48hrs post ERV stimulation (fig 6). Both CCL5, along with the IL-23 receptor (IL-23R) have both been observed to be circulating biomarkers of senescence⁷⁰. Collectively, these responses suggest that acute ERV stimulation induces a response converging on type I interferon response.

We did not observe changes to IFN- γ in our acute paradigm, although the chronic environment of the HP1 β / γ DKO brain shows several pathways that point to activation of IFN- γ and a type II interferon response. Gene set enrichment analysis of HP1 β / γ DKO RNAseq shows increases for MHC class I and II antigen presentation pathways (fig **S2f**, **Data S2**), known to be potently activated by IFN- γ ⁷¹. We also note the upregulation of IFNGR2 (**Data S1**), the regulatory subunit⁷² of the interferon gamma receptor in HP1 β / γ DKO hippocampi, suggesting greater sensitization to type II interferon signaling. In a chronic environment this may be especially problematic, given IFN- γ stimulation has been found to potentiate immune responses by initiating transcription of a subset of ERV dsRNA, which in turn triggers additional RIG-1/MDA5 signaling⁷³. Astrocytes themselves seem to be highly sensitive to IFN- γ , with effects ranging from protective responses to cytotoxicity⁷⁴. Activated astrocytes in many neurodegenerative diseases show upregulation of IFNGRs⁷⁵, suggesting an increase in sensitivity to IFN- γ signaling. Once effected, IFN- γ signaling has been observed to up-regulate expression of the complement components C3 and C4 by stabilization of their mRNA^{76,77}. Notably, in addition to other non-inflammatory roles, IFN- γ signaling has been shown to promote tau hyperphosphorylation⁷⁸.

Our results add to a growing body of work implicating reactive RNA species, complement and unfolded protein response (UPR) engagement in neurodegenerative disease; cross-talk between innate immune pathways (complement) and the unfolded protein response (UPR) is known collectively as the integrated stress response (ISR)⁷⁹. Complement is activated in TDP-43 proteinopathies⁸⁰ which can be potentiated by elevated ERV expression²². Complement C1q also forms condensates in an RNA dependent manner that can drive neurodegeneration^{81 81}, which is consistent with what we observe, namely that astrocytes in HP1DKO hippocampi accumulate C3 in an environment rich in ERV RNA (fig 5). UPR engagement in HP1 β / γ DKO hippocampi is likely a direct result of sustained ERV transcription (fig **1e**, **S2f,g**). It is also known that extended UPR stress in astrocytes results in complement activation⁶⁷. Given the majority of neurodegenerative diseases are characterized by misfolded proteins or altered proteostasis^{79,82}, our study provides an invaluable insight into how heterochromatin loss with concomitant (re)activation of ERVs can drive core components of neurodegeneration⁸³ by stimulation of immune pathways and the ISR. “

The relevance of the unfolded protein response (UPR) and complement cascade in HP1 KO mice remains ambiguous in the RNA-seq data, as these are not prominent in the top GO terms, which instead highlight cell adhesion and synaptic processes. Although identified through GSEA analysis, the specific gene sets used and the significance of

enrichment are not presented; instead, network nodes are generated with Cytoscape. Furthermore, the functional relevance of upregulation of Protocadherin genes in the HP1 γ KO has not been discussed (cell adhesion is one of the top GO terms in the RNA-seq results).

We thank the reviewer for their rigour. We think this may be because overrepresentation analysis on the relatively conservative set of 649 significantly changed genes cannot fully recover the inflammatory signature from our bulk RNAseq. We have included further plots from the GSEA data in Figure S2, and have included the table of the full GSEA results in Data S2 which show the ranking of categories associated with unfolded protein response and immune categories.

In this case we made a conscious decision not to explore the protocadherins in the HP1DKO phenotype given the pro-inflammatory effects of ERVs seemed to be the most dramatic phenotype in HP1DKO brains. We do have a separate study running that is examining protocadherin regulation in the HP1gKO mutant only. When we looked into the cell adhesion and synaptic process gene categories previously, these were heavily biased towards protocadherins and semaphorins, and specifically Sema4D. However, semaphorin signaling is also known to be important in Astrocyte-microglia communication. We have made emphasis of this in the manuscript discussion text:

“Alongside immune and complement responses, HP1 β / γ DKO hippocampi show enrichment in semaphorin signaling pathways (fig S2g, Data S2), which we attribute to known semaphorin signatures in microglia-astrocyte communication during reactive chemotaxis^{62,63}.”

Overall, the presentation of the RNA-seq data is somewhat confusing, and the selection of genes for follow-up doesn't always seem statistically driven.

We appreciate the reviewer's concern. We think that given the modification of Figure 1 to show the number of ERV loci that are de-repressed, along with the extension of figure S2, and the inclusion of the GSEA table in Data S2 the reason for our follow up genes is more obvious. We have also reworked the discussion text to be more declarative on the main importance message of this work; given we know viral RNA can drive UPR responses, and UPR responses can result in complement activation, we have observed molecular signatures ranging from endogenous viral RNA all the way to complement activation in vivo, with accompanying effects in neurodegeneration and cognitive ability.

Specific comments are as follows:

- Data S1/Table S1 lacks a list of differentially expressed genes (DEGs) and ERVs along with their expression values and q-values.

We regret this might be a CMS conversion error. Data S1 is an .xlsx with many sheets, the DEGs including expression values and FDR values are found on sheet

2., along with imprinted gene lists and interferome related lists, primers and antibodies.

- It is unclear how many endogenous retroviruses (ERVs) are significantly differentially expressed.

In order to address this we additionally performed a loci-specific analysis which is now in Figure 1b. In addition to detecting a handful of IAP loci that are de-repressed in HP1b and HP1g single mutants, we can observe a 200+ unique loci where ERV derepression takes place. We have included the table for this analysis in Data S3 which contains the raw counts and the statistics resulting from DESeq2 analysis.

- Sequencing alignment statistics for RNA-seq and ChIP-seq libraries are missing.

We have included sequencing depth and alignment efficiency for RNAseq, ChIPseq and RRBS experiments in a new sheet in Data S1.

- The ChIP-seq protocol requires a more detailed description, including any statistical analyses used to detect differentially enriched genomic regions.

We have expanded the ChIP-seq analysis to provide the full parameters of deeptools comatrix generation and heatmap plotting. We did not perform any global differential analysis to detect differentially enriched genomic regions. In this case the purpose of the ChIPseq experiment was to test regions such as IAPs and ICRs where we suspected KAP1, H3K9me3, or H4K20me3 might be affected due to HP1 loss. We look forward to performing more extensive ChIPseq analysis in future studies where we can isolate pure cell populations from the brain using reporter mouse lines.

- Some figures and figure legends need revision, for instance, Figure 1c lacks a scale bar, and several figure legends describe the results/interpretation of the data without providing accurate description of the data (e.g. Fig. s1f).

We thank the reviewer for these points to further improve the manuscript. We have included the scale bar on Fig 1c, and revised every figure caption to ensure more explicit description and have moved any minor interpretations of the data to the main text.

Glial Reactivity and Cognitive Decline Follow Chronic Heterochromatin Loss in Neurons

Authors: A.G. Newman^{1*}, J. Sharif², P. Bessa¹, S. Zaqout³, J. P. Brown^{1§}, D. Richter¹, R. Dannenberg¹, M., Nakayama⁴, S. Mueller^{5,6}, T. Schaub¹, S. Manickaraj¹, P. Böhm-Sturm^{5,6}, O. Ohara^{4,7}, H. Koseki², P.B. Singh^{1,8,#,*}, V. Tarabykin^{1,9,#,*}

RESPONSE TO REVIEWER COMMENTS – FINAL REVISION

Reviewer #3 (Remarks to the Author):

The authors have addressed several of the previous comments, including by providing clarifications in the text, expanding the discussion, incorporating GSEA plots, and missing supplementary information.

On the issue of cell type-specific ERV activation, the authors state that while HP1 proteins are deleted from both neurons and astrocytes, ERV expression is not observed in astrocytes (e.g., Fig. 5d,f). However, these panel don't clearly show astrocyte-specific co-staining with IAPs or other ERVs, and the resolution is insufficient to support this conclusion. This point should either be addressed with higher-resolution images and quantification or clearly acknowledged as a limitation, as ERV expression in astrocytes would significantly affect the overall interpretation of the findings.

We have included a new staining (Figure S7) where we have co-stained for IAP ERV RNA (probe, RNAscope) and GFAP (antibody) in the hippocampus, which confirms IAP ERV RNA upregulation is restricted to neurons.

Regarding IAP ssRNA-induced activation of microglia and astrocytes, the authors state that no C3 changes were detected by Western blot at 24 or 48 hours ("data not shown"). These data should be included in supplementary information, and if the model doesn't recapitulate the in vivo environment, this should be noted as a limitation.

We have included this western blot in the supplemental information and have further discussed the C3 response we observe in vivo cannot be recapitulated in vitro. Additionally, we highlight the fact that C3 is accumulated in the endfeet of astrocytes, "...which contact endothelial cells and pericytes of the blood brain barrier, [and] suggests part of the complement response to ERV RNA in the HP1 β / γ DKO brain may involve a greater systemic immune response."

The ChIP-seq experimental methods section remains limited. Inspection of the GEO dataset suggests no Input control was used, which limits confidence in differential enrichment results. This omission should be acknowledged as a limitation in the manuscript.

We have included the following statement in the results regarding the ChIPseq data from HP1cTKO cells:

“We acknowledge that these ChIP-seq results did not have input normalization, but have been normalized using 1× RPGC (Reads Per Genomic Content) to account for sequencing depth and genome mapability.”

Finally, Fig. 5n (iii) appears out of focus and should be reviewed or replaced.

We have replaced Figure 5n (iii), our airyscan deconvolution had failed for this image, we thank the reviewer for catching this error.

Overall, while some points have been addressed, some claims, particularly regarding cell type specificity and C3 activation, require either additional data or clearer acknowledgment of limitations.

We have made these statements in the manuscript:

In the results:

“Neither our 24hr or 48hr paradigms could detect changes in cytoplasmic C3 by Western blot (see Supplemental Information), however, our acute experimental protocol is unlikely to fully recapitulate the chronic inflammatory environment in the HP1β/γDKO brain.”

In the discussion:

“When we tested the effect of non cell-autonomous ERV RNA on mixed glial cultures, we could observe some ERV RNA specific effects. The acute ERV ssRNA stimulation was followed by IRF7 phosphorylation and nuclear ingress (fig 6e-g). IRF7 activation occurs downstream of Pattern Recognition Receptors (PRRs) that recognize ssRNA, either endosomal toll receptors TLR7/9, or cytoplasmic receptors such as RIG-1/MDA5⁶⁶. Once in the nucleus, phosphorylated IRF7 is known to directly activate type I interferons (IFNα/β) and Interferon Stimulatory Genes (ISGs) ⁶⁷, including the cytokines CXCL10 and CCL5⁴⁶, which we robustly observe 24hrs and 48hrs post ERV stimulation (fig 6). Both CCL5, along with the IL-23 receptor (IL-23R) have both been observed to be circulating biomarkers of senescence⁶⁸.

Collectively, these responses suggest that acute ERV stimulation induces a response converging on type I interferon response.

We did not observe changes to IFN- γ (type II interferon) in our acute paradigm, although the chronic environment of the HP1 β / γ DKO brain shows several pathways that point to activation of IFN- γ and a type II interferon response. Gene set enrichment analysis of HP1 β / γ DKO RNAseq shows increases for MHC class I and II antigen presentation pathways (fig S2f, Data S3), known to be potently activated by IFN- γ ⁶⁹. We also note the upregulation of IFNGR2 in HP1 β / γ DKO hippocampi (Data S1), the regulatory subunit⁷⁰ of the interferon gamma receptor, suggesting greater sensitization to type II interferon signaling. In a chronic environment this may be especially problematic, given IFN- γ stimulation has been found to potentiate immune responses by initiating transcription of a subset of ERV dsRNA, which in turn triggers additional RIG-1/MDA5 signaling⁷¹. Astrocytes themselves seem to be highly sensitive to IFN- γ , with effects ranging from protective responses to cytotoxicity⁷². Activated astrocytes in many neurodegenerative diseases show upregulation of IFNGRs⁷³, suggesting an increase in sensitivity to IFN- γ signaling. Once effected, IFN- γ signaling has been observed to up-regulate expression of the complement components C3 and C4 by stabilization of their mRNA^{74,75}. Notably, in addition to other non-inflammatory roles, IFN- γ signaling has been shown to promote tau hyperphosphorylation⁷⁶. In summary, we have observed that acute paradigms of ERV stimulation can robustly activate cytokines consistent with a type I interferon response, the chronic environment of the HP1 β / γ DKO brain suggests chronic activation of type II interferon response, which has several known synergies with complement signaling.”